EMBO
Molecular Medicine

# IGFBP3 repression driven by inflammation links air pollution to placental and developmental defects

Sunil Singh [ID][1], Isha Goel[1,2], Anubhuti Rana[3], Anamta Gul [ID][1], Javed A Quadri [ID][4], Asit Ranjan Mridha [ID][5], Lakshay Malhotra[6], Neha Kashyap[1], Baburajan Radha[1], Arnab Nayek [ID][1], Swati Ajmeriya[1], Jitender Prasad[7], Ruby Dhar [ID][1✉] & Subhradip Karmakar [ID][1✉]

## Abstract

**Air particulate matter (PM2.5 and PM10), can cross the placental barrier, triggering oxidative stress and inflammation that compromise fetal development. These insults lead to placental dysfunction and complications including preterm birth, low birth weight, and preeclampsia. In cell line and placental explant models, urban particulate matter (UPM) increased pro-inflammatory cytokines and oxidative stress pathways, impairing trophoblast invasion, angiogenesis, and nutrient transport, while also altering epigenetic modifications and endoplasmic reticulum function. Rodent studies revealed reduced litter size, placental abnormalities, and fetal growth arrest along with postnatal neurodevelopmental alterations. Human cohorts from high-exposure regions showed elevated low birth weight rates. Proteomic and transcriptomic analyses of rat placenta revealed an inflammatory signature and altered metabolic networks, while gut microbiome dysbiosis suggested links to metabolic disturbances. Importantly, transcriptomic analysis identified IGFBP3 as a major downregulated gene following UPM exposure. IGFBP3, a key regulator of IGF bioavailability, was suppressed by IL1β, establishing inflammation-driven repression as the mechanism. These findings underscore UPM's multidimensional impact on maternal–fetal health and highlight preventive strategies as urgent priorities.**

**Keywords** Urban Particulate Matter; Trophoblast differentiation; Inflammation; Insulin-like Growth Factor Binding Protein 3 (IGFBP3); Fetal Neurodevelopment
**Subject Categories** Development; Evolution & Ecology; Immunology

## Introduction

Particulate matter (PM) is a harmful air pollutant and has become a global health concern. These can be classified as PM0.5, PM2.5 and PM10 (Pryor et al, 2022). The toxicity of PM relates to their composition, quantity, mass, shape, size, and surface area. PM of diameter <10 μm have the potential to enter the bloodstream, leading to premature deaths in vulnerable populations with co-morbidities like heart or lung diseases, diabetes mellitus, malnutrition, and hypertension (Li et al, 2022; Bennitt et al, 2025). Recent research has shown links between air quality and adverse pregnancy outcomes, suggesting air pollutants disrupt processes involved in placentation and maternal-neonatal health (Singh et al, 2024; Soesanti et al, 2023; Mandal et al, 2023). The placenta is a transient yet crucial organ that delivers oxygen and nutrients to the fetus while disposing of waste materials. The placenta performs vital endocrine functions during pregnancy, synthesizing hormones that support fetal development and maternal physiological adaptations. It also exhibits metabolic plasticity, enabling it to respond to environmental changes and meet the dynamic needs of both mother and fetus (Burton and Jauniaux, 2023; Ortega et al, 2022).

However, studies have shown that certain environmental pollutants can breach the placental barrier, impairing its function, consequently affecting maternal and fetal health (Bové et al, 2019; Zurub et al, 2024). Placental cells consisting of an invasive cytotrophoblast and fusogenic syncytiotrophoblast, have been shown to be affected by pollutants (Knöfler et al, 2019). A landmark study by Bongaerts et al demonstrated the presence of black carbon particulates from polluted air in cord blood, indicating that air pollutants can breach the placental barrier and reach the fetus, contributing to harmful effects (Bongaerts et al, 2022). In an urban Tanzanian cohort with personal exposure monitoring, Wylie et al observed a 0.15 kg decrease in birth weight per interquartile increase (23.0 μg/m³) in maternal PM2.5 exposure, (95% CI: −0.30 to 0.00 kg; *p* = 0.05) (Wylie et al, 2017). Although modest in magnitude, a 100–150 g left-shift is clinically meaningful at the population level: by WHO/GAIA (World Health Organization/Global Alignment of Immunization Safety Assessment in Pregnancy) criteria, LBW is <2500 g, and such shifts increase the share of infants crossing the LBW threshold—an outcome linked to markedly higher neonatal mortality and long-term neurodevelopmental and cardiometabolic risks—especially in regions with already high LBW prevalence(Brämer, 1988). Multiple

[1]Department of Biochemistry, All India Institute of Medical Sciences, New Delhi, India. [2]Department of Laboratory Medicine, All India Institute of Medical Sciences, New Delhi, India. [3]Department of Obstetrics and Gynaecology, All India Institute of Medical Sciences, New Delhi, India. [4]Department of Anatomy, All India Institute of Medical Sciences, New Delhi, India. [5]Department of Pathology, All India Institute of Medical Sciences, New Delhi, India. [6]Department of Biochemistry, Sri. Venkateswara College, University of Delhi, New Delhi, India. [7]Department of Biochemistry, All India Institute of Medical Sciences, Deoghar, Jharkhand, India. ✉E-mail: rubydhar@aiims.edu; subhradip.k@aiims.edu

epidemiological studies across diverse populations have established a strong association between PM2.5, PM10 exposure and adverse pregnancy outcomes. Pope et al reported pooled odds ratios of 1.13–1.87 for LBW due to indoor air pollution across India, Guatemala, Zimbabwe, the USA, and Pakistan (Pope et al, 2010). In India, where solid fuels remain a major household energy source, household air pollution continues to contribute significantly to adverse pregnancy outcomes (Gupta et al, 2014; Kaur and Pandey, 2021; Mukherjee et al, 2021). Amegah et al showed that solid-fuel exposure led to an 86.4 g reduction in birth weight (95% CI: 55.5–117.4) and a 35% increased LBW risk (Amegah et al, 2014). Similarly, Capobussi et al reported adverse pregnancy outcomes with $SO_2$ (aOR = 1.24) and PM10 (aOR = 1.07) exposure in Italy (Capobussi et al, 2016), while in the United States, an analysis of 32.8 million births indicated that 25 of 29 studies (2007–2019) reported a positive association between ozone or PM2.5 exposure and LBW, with stillbirth risk increasing up to 42% during the third trimester (Bekkar et al, 2020; DeFranco et al, 2015). A European meta-analysis further confirmed that a 10 µg/m³ increase in PM2.5 exposure elevated LBW risk by 1.39–1.98-fold, reinforcing the global, consistent, and dose-dependent association between PM2.5, PM10 exposure and impaired fetal growth (Simoncic et al, 2020). Additional studies by X Li et al, Ghosh et al, Gong et al, and Younger et al reported a positive correlation between indoor and outdoor air pollutants—including nitrogen oxide (NO), carbon monoxide (CO), sulfur dioxide ($SO_2$), ozone ($O_3$), PM2.5, and PM10—and the incidence of low birth weight (LBW) (Li et al, 2017; Ghosh et al, 2021; Gong et al, 2022; Younger et al, 2022). There is also a strong association between PM2.5, PM10 exposure and pregnancy complications such as preeclampsia, intrauterine growth restriction, miscarriage, and gestational diabetes mellitus (Singh et al, 2024).

Emerging mechanistic evidence implicates KLF9/CYP1A1-directed oxidative stress and mitochondrial apoptosis in mediating PM2.5 effects on placental function (Li et al, 2023). Specifically, PM2.5-associated polycyclic aromatic hydrocarbons (PAHs) induce placental cytochrome P450 1A1 (CYP1A1)—a xenobiotic-metabolizing enzyme that, while functioning to detoxify environmental pollutants, paradoxically generates reactive oxygen species (ROS) as metabolic byproducts (Whyatt, 1998; Vogel et al, 2020). These ROS overwhelm trophoblast antioxidant defenses, triggering oxidative damage to cellular macromolecules while activating the intrinsic mitochondrial apoptotic pathway through cytochrome c release and caspase activation (Wu et al, 2015). Trophoblasts—the specialized placental cells mediating maternal-fetal nutrient exchange, hormone synthesis, and immune regulation—are particularly vulnerable to oxidative injury due to their high metabolic demand and direct exposure to maternal circulation (Poston and Raijmakers, 2004). Excessive trophoblast apoptosis reduces functional placental mass, impairs nutrient/oxygen transfer capacity, and compromises endocrine support for fetal growth, thereby leading to adverse birth outcomes (Poston and Raijmakers, 2004; Straszewski-Chavez et al, 2005). Krüppel-like factor 9 (KLF9), is a transcription factor that regulates oxidative stress responses and has been shown to mediate mitochondrial homeostasis in placental cells (Li and Chen, 2025). PM2.5-induced dysregulation of KLF9 appears to mediate these pathological responses, with altered KLF9 expression disrupting key placental developmental programs (Li and Chen, 2025). KLF9 is explored for its role in preeclampsia,

where it is thought to mediate oxidative stress and inflammation in trophoblasts (Zucker et al, 2014; Li and Chen, 2025; Li et al, 2023). This factor may suppress the antioxidant gene PRDX6, leading to an imbalance in ROS and the activation of the NLRP3 inflammasome, which releases inflammatory cytokines linked to preeclampsia pathology (Li and Chen, 2025).

A recent report by Shuxian Li et al, identified KLF9/CYP1A1-directed trophoblast oxidative stress and mitochondrial apoptosis as an underlying mechanism for PM-induced adverse pregnancy outcomes (Li et al, 2023). In mouse models, exposure to diesel exhaust particles (DEP) has been associated with an increase in inflammatory markers such as IL-2, IL-5, IL-12α, IL-12β, and GM-CSF (Alvarez-Simón et al, 2017; Daniel et al, 2021). These adverse effects are not limited to mothers; they also impact offspring. The Developmental Origins of Health and Disease (DOHaD) framework suggests that prenatal exposure to hazardous pollutants can lead to adverse birth outcomes, alter fetal physiology, disrupt metabolic processes, and increase the risk of chronic conditions later in life (Lacagnina, 2020; Barker, 1986; Langley-Evans and McMullen, 2010).

Although numerous studies associate PM2.5 exposure with adverse pregnancy outcomes, the molecular pathways linking placental dysfunction to inflammatory stress remain poorly defined. Increasing evidence shows that fine particulate matter (PM2.5) and its PAH constituents elicit endoplasmic reticulum (ER) stress, a central mechanism coupling oxidative injury to inflammation (Heng et al, 2021; Kim et al, 2024; Du et al, 2022). In trophoblast cultures, PM2.5 exposure markedly upregulated ER stress, elevated ROS production, and induced apoptosis accompanied by increased IL-1β and TNF-α secretion (Lee et al, 2019; Pu et al, 2023; Heng et al, 2021; Familari et al, 2019). Similar findings in bronchial and hepatic models demonstrate that ER stress sensors drive NF-κB activation, amplifying cytokine release (Yeong and Chew, 2016; Pu et al, 2023). In the placenta, persistent activation of the ER stress axis impairs trophoblast invasion, suppresses Matrix Metalloproteinase-2 (MMP-2), and disrupts angiogenic signaling, contributing to preeclampsia-like pathology (Capatina et al, 2021; Burton et al, 2017). Considering these findings and the knowledge gap, we hypothesized that UPM exposure initiates ER stress as a primary molecular trigger, integrating oxidative and inflammatory pathways that compromise trophoblast differentiation and immune equilibrium at the maternal–fetal interface. The resulting ER stress–driven cytokine surge and unfolded-protein-response signaling perpetuate chronic inflammation, leading to placental insufficiency and other pregnancy complications. This perturbation thus can be attributed to adverse neurobehavioral and other clinical outcomes in fetus. By defining ER stress-mediated inflammatory signaling as the pivotal mechanism of UPM toxicity, this study advances a novel framework linking environmental particulate exposure to disrupted placental function and developmental outcomes.

# Results

## Urban particulate matter exposure inhibits trophoblast invasion and syncytialization

To investigate the effects of UPM exposure on invasion and migration of HTR8/SVneo cells, we assessed the expression levels of

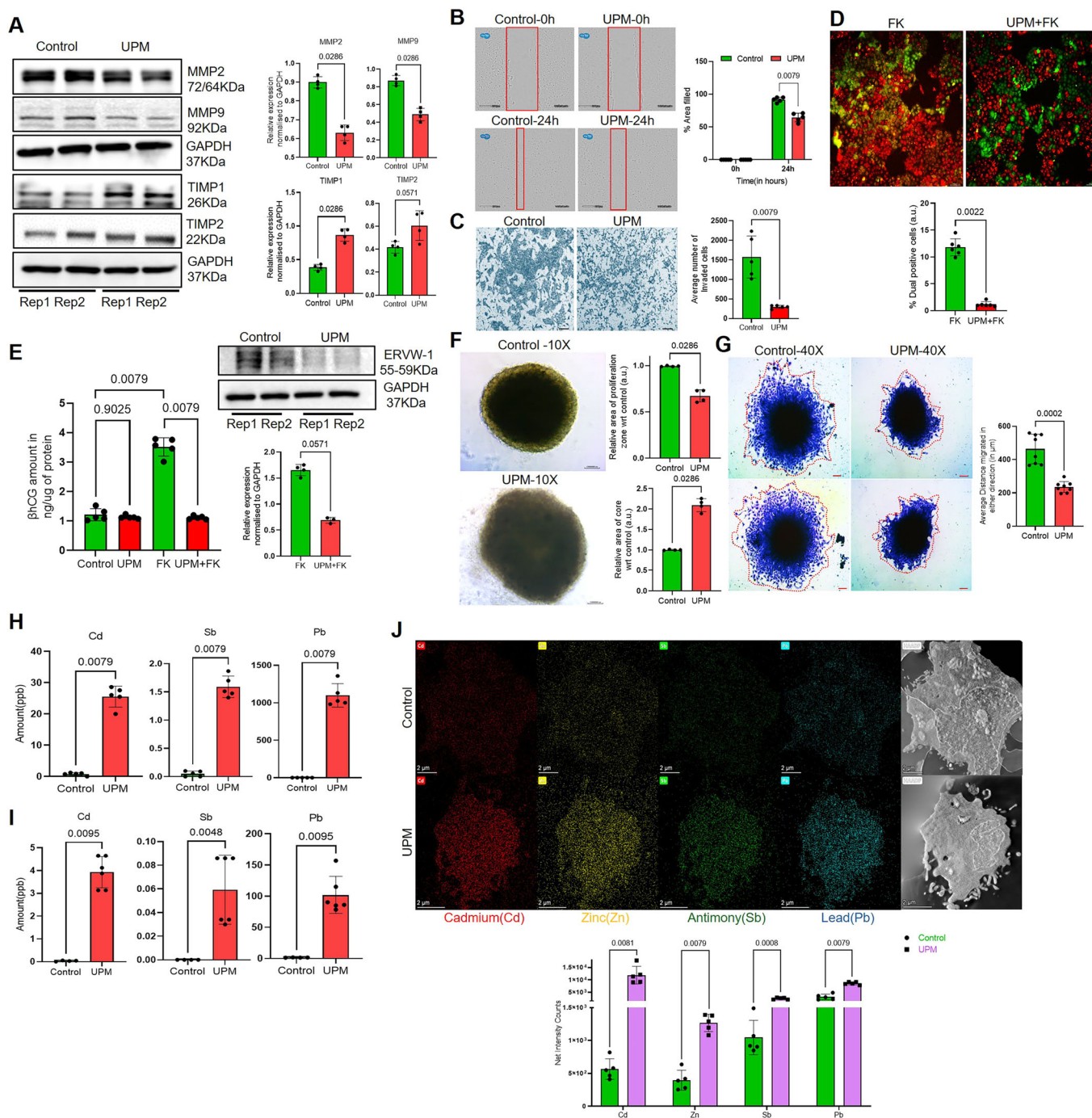

matrix metalloproteinases MMP2 and MMP9, as well as the tissue inhibitors of metalloproteinases TIMP1 and TIMP2 following 50 µg/mL of UPM treatment (Appendix Fig. S1). We noted a significant reduction in the expression of MMP2 (approximately 1.5-fold, $p < 0.028$) and MMP9 (approximately 1.8-fold, $p < 0.028$), with upregulated TIMP1 (approximately 2.2-fold, $p < 0.028$) and TIMP2 (approximately 1.5-fold, $p < 0.057$) expression in the UPM-exposed cells (Fig. 1A). Gelatin zymography from spent media of UPM treated HTR8/SVneo cells showed reduced gelatinolytic bands implying reduced MMP production or enzymatic activity

(Fig. EV1A). Migration analyses showed impaired trophoblast motility, with UPM-treated cells achieving $61 \pm 3\%$ wound closure versus $89.7 \pm 4\%$ in controls (Fig. 1B). Invasion analysis revealed reduced trans-membrane invasion, with UPM-exposed trophoblasts showing $240 \pm 56$ cells compared to $1500 \pm 600$ control cells (Fig. 1C), indicating compromised cellular invasion behaviors affecting early pregnancy processes.

The syncytiotrophoblast performs vital functions, including βhCG hormone biosynthesis and materno-fetal substrate transport across the placental interface. Forskolin-induced cAMP elevation

**Figure 1. UPM negatively affects the invasion and syncytialization of trophoblast cells.**

(A) Western blot showed reduced expression of MMP2 and MMP9, while the expression of TIMP1 and TIMP2 was upregulated following UPM stimulation. Band intensities were quantified using ImageJ and normalized to the GAPDH (housekeeping). Rep1-Rep2: 2 biological replicates ($n = 4$). (B) UPM-exposed cells exhibited significantly diminished wound closure rates at 24 h compared to untreated control. Quantitative analysis involved percentage wound area closure at baseline (0 h) and 24 h timepoints. Wound area quantification was performed using Cytoscape imaging software ($n = 5$). Scale bar equals 200 μm. (C) A significantly reduced number of cells was observed at the bottom side of Matrigel membranes in the UPM group compared to the control group ($n = 5$). Scale bar equals 200 μm. (D) Fluorescence microscopy analysis revealed a substantial decrease in syncytial cell formation following UPM treatment, even when forskolin was co-administered to stimulate fusion. The fusion index was quantitatively determined by calculating the proportion of yellow fluorescent regions relative to total cellular area. Data analysis focused on comparing the percentage of dual-fluorescence positive (yellow) syncytial zones between the UPM + FK treatment group and the FK-only control group ($n = 6$). Scale bar equals 100 μm. (E) βhCG-ELISA ($n = 5$) and western blotting for Syncytin-1 (ERVW-1) ($n = 4$) showed a significant reduction ($p = 0.0079$) after UPM stimulation, even in the presence of FK. (F) UPM stimulation of spheroids resulted in diminished proliferative zones and expanded necrotic core regions. Statistical significance was determined with sample size $n = 4$. (G) Cellular egress (marked as red dotted boundary) from UPM-treated spheroids demonstrated significant impairment compared to control spheroids. Quantitative analysis was done for cell dispersion in all radial directions from the spheroid periphery, with data subsequently analyzed and presented graphically. Statistical significance was determined with sample size $n \geq 5$. (H, I) ICP-MS analysis revealed significantly elevated concentrations of cadmium (Cd), antimony (Sb), and lead (Pb) in both HTR8/SVneo (H) and BeWo cells (I) following UPM exposure. Statistical significance was determined with sample size $n = 5$. (J) STEM-EDS spectra revealed significantly higher signals for Cd ($p = 0.0081$), Sb ($p = 0.0008$), Pb ($p = 0.0003$) and Zinc (Zn) ($p = 0.0079$) in UPM-treated cells compared to untreated controls. Net intensity plots were generated from STEM-EDS data to quantify metal enrichment within the cells ($n = 5$). Scale bar equals 2 μm. All data are presented as mean ± SD. Statistical significance was determined using two-tailed Mann-Whitney test. Source data are available online for this figure.

triggers fusogenic differentiation in BeWo choriocarcinoma cells in vitro (Renaud and Jeyarajah, 2022). Fusion assays with CMTMR-Orange and CMFDA-Green co-stained BeWo cells showed that forskolin promotes membrane fusion, visible as yellow fluorescent syncytia. UPM co-treatment suppressed syncytial formation by 7.5-fold compared to forskolin alone, indicating disrupted syncytialization (Fig. 1D). ELISA confirmed that forskolin-induced syncytialization correlates with increased βhCG production which was reduced 3.5-fold in the presence of UPM (Fig. 1E). qPCR and Immunoblot analysis confirmed downregulation of fusion mediating syncytin-1 (2.5-fold, $p < 0.0571$) protein with UPM exposure (Figs. 1E and EV1B–D). These analyses demonstrate that UPM exposure disrupts trophoblast syncytialization through suppression of fusion-regulatory pathways, potentially compromising placental function.

To eliminate potential confounding from two-dimensional culture artifacts, we employed three-dimensional spheroid models to validate our observations. These models better recapitulate native tissue architecture and microenvironments compared to monolayer systems. Mature spheroids develop distinct spatial compartmentalization with a hypoxic/necrotic core surrounded by an actively proliferating periphery, mimicking natural tissue organization (Stojanovska et al, 2022). HTR8/SVneo spheroids exposed to UPM exhibited aberrant morphology, with disrupted architecture, poorly defined core-periphery boundaries, and compromised proliferation zones, while vehicle-treated controls maintained characteristic spherical geometry and clear compartmentalization (Fig. 1F). Quantitative analysis showed significantly reduced cellular outgrowth and migratory capacity in UPM-treated spheroids (Fig. 1G).

Ex vivo validation from first-trimester villi and term placenta explants confirmed that UPM exposure induces significant dysregulation across critical placental pathways. UPM treatment impaired invasiveness by downregulating MMP2 and MMP9 while upregulating TIMP1, and compromised syncytialization as evidenced by reduced Syncytin-1, βhCG, and GCM1 expression (Appendix Fig. S2A,B). Concurrently, UPM triggered a robust inflammatory response, significantly upregulating IL-1β, IFN-α, IFN-γ, IL-2, and TGF-β transcripts (Appendix Fig. S2C,D). These findings indicate that UPM disrupts placental functions

through coordinated suppression of invasion and fusion pathways alongside inflammatory activation. Overall, these in vitro, ex vivo and three-dimensional validations confirm that gestational UPM exposure compromises placental morphogenesis by disrupting trophoblast proliferation, organization, and invasion.

To explore the possibility of accumulation of metals from UPM, in stimulated trophoblast we performed Inductively Coupled Plasma–Mass Spectrometry (ICP-MS). Results showed significant intracellular accumulation of toxic metals following UPM exposure. Our findings revealed a substantial increase in cadmium (Cd), antimony (Sb), and lead (Pb) concentrations in trophoblast cells (Fig. 1H,I; Appendix Fig. S3A–C). Transmission and Scanning Transmission Electron Microscopy (TEM/STEM) Analysis of UPM-Exposed HTR8/SVneo cells was also done and data indicated prominent mitochondrial changes with high signals for Cd, Pb and Sb (Figs. 1J and EV1E). This metal ion bioaccumulation suggests that UPM exposure could deliver and accumulate toxic metals into placental tissue thereby demonstrating the impact of heavy metals in UPM on trophoblast function.

## Transcriptomic analysis reveals UPM-induced inflammatory reprogramming in trophoblast systems

A total of 8 RNA samples comprising of 2 EP-Control, 2 EP-UPM treated, 2 HTR8/SVneo control and 2 UPM treated HTR8/SVneo cells were used. The sequencing was performed on an Illumina NovaSeq 6000 V1.5 platform using the cDNA library. Paired end sequencing ($2 \times 150$ bp) was carried out to achieve 50 million reads from each sample. To elucidate molecular mechanisms underlying UPM-mediated trophoblast dysfunction, transcriptome profiling identified 27,153 expressed genes in placental explants and 22,316 genes in HTR8/SVneo cells, with principal component analysis implying a clear sample clustering indicating robust transcriptional reprogramming following UPM exposure (Appendix Fig. S4). Differential expression analysis using stringent criteria (padj<0.05, | log2FC| ≥1) revealed 196 dysregulated genes in HTR8/SVneo cells (71 upregulated, 125 downregulated) (Appendix Fig. S5) and 45 differentially expressed genes in first-trimester explants (20 upregulated, 25 downregulated). Volcano plot and heatmap

visualizations confirmed distinct expression signatures between control and UPM-treated conditions (Appendix Fig. S4).

Gene set enrichment analysis identified critical pathway perturbations (Fig. 2A,B), with inflammatory cascades including IFN-γ-mediated responses, TNF signaling, reactive oxygen species generation, and xenobiotic metabolism pathways significantly enriched in UPM-exposed samples. Conversely, essential developmental processes including angiogenesis and epithelial-mesenchymal transition were substantially downregulated (Fig. 2C), indicating compromised trophoblast differentiation capacity. Explant-specific pathway analysis revealed elevated IL-6 signaling, inflammatory responses, IFN-γ activation, and hypoxic stress pathways.

These transcriptional alterations suggest UPM exposure has the potential to disrupt the delicate immunological balance required for successful placentation. Physiologically, placental IL-4 and IL-10 production promotes protective Th2-type immune responses, while UPM-induced IL-6 upregulation may trigger detrimental Th1-mediated inflammation. Enhanced IFN-γ signaling activates JAK/STAT pathways that suppress extravillous trophoblast invasion through STAT1-mediated BATF2 downregulation and promotes apoptotic cell death. UPM exposure initiates maladaptive inflammatory reprogramming, compromising placental functions including invasion, syncytialization, and vascularization, potentially contributing to adverse pregnancy outcomes.

## UPM-mediated trophoblast dysfunction through endoplasmic reticulum stress

RNA sequencing data revealed exposure to particulate matter induces cytotoxic mechanisms, including apoptotic cascades, oxidative stress, and endoplasmic reticulum (ER) stress. Based on our transcriptomic findings, we investigated whether UPM-mediated cellular dysfunction results from ER stress. Immunoblot analysis of ER stress markers XBP1s, CHOP, BiP, and IRE1α in HTR8/SVneo (Fig. 3A) and BeWo cells (Fig. 3C), along with immunofluorescence localization of XBP1s in HTR8/SVneo (Fig. 3B), revealed significant upregulation of unfolded protein response components following UPM exposure compared to controls. To establish mechanistic causality, N-acetylcysteine (5 mM) was used as a cytoprotective agent during UPM treatment in BeWo cells (Fig. 3C).

Given epidemiological evidence linking air pollution to epigenetic changes affecting pregnancy, we conducted a preliminary investigation of UPM-induced chromatin alterations. Histone H3 modification analysis in HTR8/SVneo cells showed significant Histone-3 epigenetic changes following UPM exposure compared to controls (Fig. EV2A). These findings establish a mechanistic framework where UPM triggers ER stress activation and epigenetic remodeling, collectively impairing trophoblast functions critical for pregnancy establishment and maintenance.

## In vivo validation of UPM-mediated reproductive toxicity and clinical correlation with air pollution exposure

To establish physiological relevance beyond cellular and ex vivo models, we implemented a rodent exposure paradigm to assess gestational urban particle effects on maternal-fetal outcomes. Pregnant Wistar rats underwent controlled UPM exposure via intranasal route for three weeks pre-conception through parturition, with exposure concentrations calibrated to PM2.5 levels as documented for human exposure in pregnant women from New Delhi, India during 2022–24. Systematic evaluation was conducted at critical gestational timepoints (GD 7.5, 16.5, and 21.5) with comprehensive assessment of reproductive parameters including litter size, placental morphometrics (weight, diameter), and fetal biometric indices (weight, crown-rump length) (Fig. 4A).

While we did not observe any difference between maternal weight (Appendix Fig. S6A) and feeding behavior, UPM exposure significantly compromised reproductive outcomes in a gestational age-dependent manner. Litter size reduction was evident from early gestation, with progressive worsening from 12.5% reduction at GD 16.5 to 25% decrease by GD 21.5 (Fig. 4B; Appendix S6B), likely attributable to either low implantation or an increased resorption rate. Placental development was substantially impaired, with significant reductions in both placental weight and diameter observed at mid- and late gestation timepoints (GD 16.5, 21.5) (Fig. 4C). These placental growth restrictions logically suggest compromised placentation processes which may impair maternal-fetal nutrient exchange and vascular development, consistent with our in vitro findings of reduced trophoblast invasive capacity.

Fetal growth assessment revealed dose-dependent developmental impairment, with progressive reduction in both fetal weight and axial length measurements (Fig. 4D; Appendix S6B). Specifically, fetal weight decreased by approximately 9% at GD 16.5, escalating dramatically to 34% reduction by term (GD 21.5). These parameters serve as sensitive indicators of intrauterine growth restriction and reflect compromised placental-fetal communication networks essential for optimal developmental programming. Exposure to PM during pregnancy significantly alters maternal serum levels of key hormones associated with pregnancy, which has serious consequences for placental hormone function and the maintenance of pregnancy.

The analysis of maternal serum samples using ELISA showed notable decreases in CGα levels in dams exposed to UPM in comparison to control groups, with levels dropping by about 10.7% ($p < 0.019$) at late gestation (GD 21.5) (Fig. 4E). Likewise, pregnancy-associated plasma protein A (PAPP-A) exhibited considerable suppression after exposure to UPM, with serum levels declining by 48% at GD21.5, respectively (Fig. 4E). These hormonal changes are directly linked to our in vitro observations of impaired syncytiotrophoblast development and decreased β-hCG production in BeWo cells treated with UPM, demonstrating consistent mechanisms across various experimental setups. The simultaneous reduction in both CGα and PAPP-A implies a widespread disruption of syncytiotrophoblast hormone function, as these hormones are mainly produced by the multinucleated syncytial layer and play vital roles in sustaining pregnancy.

To establish clinical relevance in relation to humans, we conducted an epidemiological analysis that correlated ambient PM2.5 concentrations with pregnancy outcomes in New Delhi, India, where the annual average concentrations reached 100.9 μg/m³ in 2022–24, and in Deoghar, Jharkhand, India, where the annual average concentration was below 50 μg/m³. A total of 994 records were assessed after samples passed exclusion and inclusion criteria. Population stratification across exposure gradients ($< 50$ μg/m³, 60–100 μg/m³, $>100$ μg/m³) revealed dose-dependent decreases in

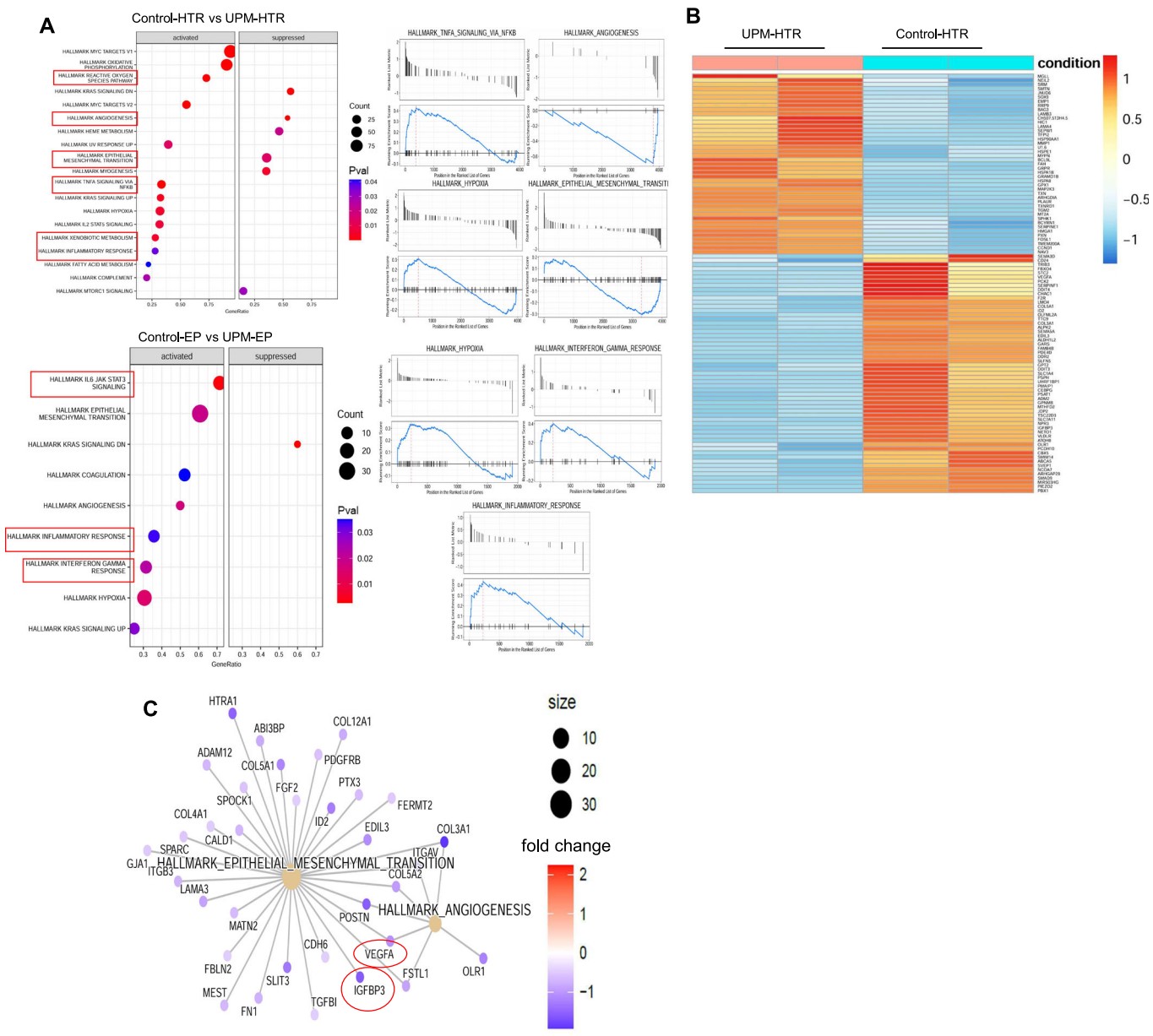

**Figure 2. Transcriptomic analysis of HTR8/SVneo cells and early placental villous explants reveals UPM-induced inflammatory and metabolic reprogramming.**

(A) Bioinformatics analysis of differentially expressed gene sets in urban particulate matter-exposed HTR8/SVneo trophoblast cells and primary early placental villous tissue explants revealed significant enrichment of multiple critical cellular pathways. Gene set enrichment analysis identified substantial upregulation of inflammatory cascade signatures, particularly cytokine-mediated signaling networks, pro-angiogenic gene expression programs, epithelial-to-mesenchymal transition regulatory pathways, hypoxia-responsive transcriptional modules, and xenobiotic detoxification metabolic processes. These findings demonstrate comprehensive transcriptional reprogramming across diverse biological systems, indicating that UPM exposure triggers widespread cellular stress responses and adaptive mechanisms within placental tissue models. (B) The heatmap visualization displays normalized expression profiles of the most significantly altered genes in control versus UPM-exposed HTR8/SVneo cells. Unsupervised hierarchical clustering revealed distinct transcriptional signatures, with clearly delineated gene clusters demonstrating divergent expression patterns in inflammatory response networks and developmental regulatory pathways following urban particulate matter treatment. These clustering patterns highlight the systematic transcriptional reorganization induced by UPM exposure. (C) The concept network (Cnet) plot illustrates the complex molecular interaction landscape of genes functionally associated with epithelial-to-mesenchymal transition and angiogenic signaling cascades. Node dimensions correlate with gene connectivity metrics (degree centrality), while color gradient intensity represents relative expression magnitude. This network topology analysis demonstrates how UPM exposure disrupts trophoblast cellular plasticity mechanisms and compromises vascular remodeling transcriptional programs, revealing the interconnected nature of perturbed biological processes.

birth weights for the incidence of low birth weight. PM2.5 was identified as a significant risk factor for low birth weight incidences (Relative Risk-2.033, CI 1.38–3.0, $p = 0.00068$). The incidence of preeclampsia showed a three-fold increase from 1.92% (PM2.5 60–80 μg/m³) to 6% (> 100 μg/m³) across the exposure spectrum, while birth weight parameters exhibited significant inverse correlations with PM2.5 concentrations, affecting both normal and low birth weight categories (Fig. 4F; Appendix Fig. S7).

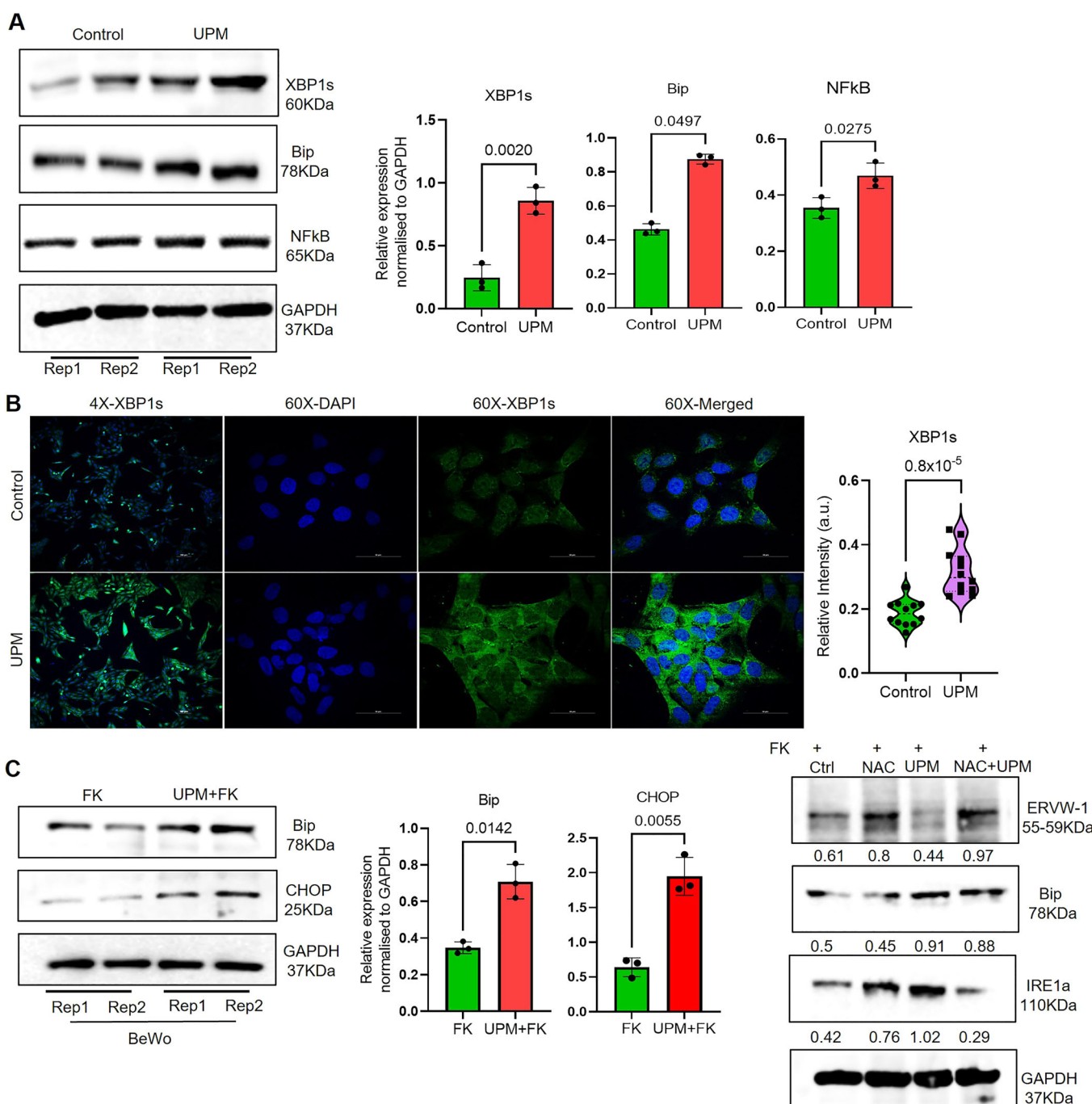

**Figure 3. UPM-mediated adverse effects on trophoblast function involve an induction of aberrant endoplasmic reticulum stress response.**

(A) Western blot analysis targeting key ER stress biomarkers, including X-box binding protein 1 spliced (XBP1s), binding immunoglobulin protein (BiP), and nuclear factor kappa B (NFκB), demonstrated substantial upregulation of unfolded protein response in UPM-treated cells relative to vehicle-treated controls. Densitometric quantification of protein bands was performed using ImageJ software, with data normalized to glyceraldehyde-3-phosphate dehydrogenase (GAPDH) loading control and presented graphically ($n = 3$, Student's t-test). (B) Confocal immunofluorescence imaging of XBP1s localization in HTR8/SVneo cells corroborated the Western blot findings. Nucleus was stained with 4′,6-diamidino-2-phenylindole (DAPI) counterstaining, while XBP1s protein distribution was detected via green fluorescence immunostaining. Enhanced fluorescence intensity was observed in UPM-exposed cells compared to controls, with quantitative analysis graphically displayed ($n = 4$, $p = 0.8 \times 10^{-5}$). Scale bar equals 100 μm in 4x and 50 μm in 60x. Statistical significance was determined using two-tailed Mann–Whitney test. (C) UPM treatment of BeWo choriocarcinoma cells resulted in pronounced elevation of ER stress markers BiP and C/EBP homologous protein (CHOP) relative to untreated controls ($n = 3$, Student's t-test). Protein band densitometry was performed using ImageJ software with GAPDH normalization, and results were graphically presented. All data are presented as mean ± SD. Co-treatment with N-acetylcysteine (NAC), a potent antioxidant, effectively attenuated UPM-induced expression of ER stress proteins BiP and inositol-requiring IRE1α. Additionally, NAC supplementation restored ERVW1 expression levels in the presence of UPM. Source data are available online for this figure.

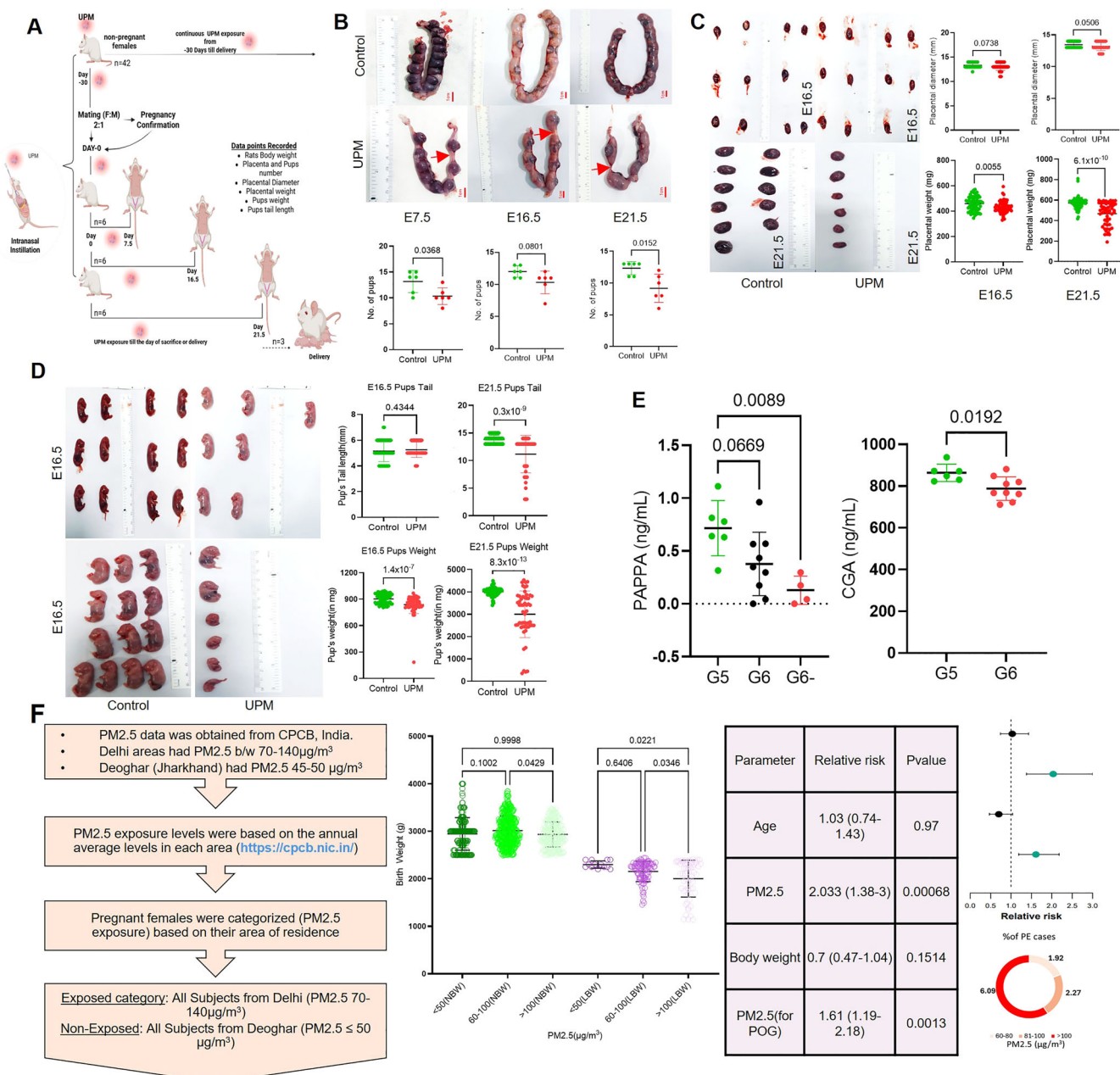

**Figure 4. Gestational exposure to particulate matter is associated with adverse pregnancy and fetal outcomes.**

(A) Diagrammatic illustration depicting the exposure protocol for UPM administration (B) UPM exposure induced progressive impairment of fetal growth across gestation. Macroscopic examination of uterine horns at E7.5, E16.5, and E21.5 showed intrauterine growth restriction and increased embryonic resorption (red arrows) in UPM-exposed dams compared with controls. Fetal numbers were significantly reduced in UPM group across the gestation (n = 6 dams per stage). (C) Placental tissue collected at GD16.5 and GD21.5 exhibited significantly reduced wet weight and diameter in UPM-exposed pregnancies compared with controls. Representative images from each group are shown alongside quantitative analyses. (D) Fetuses harvested at GD16.5 and GD21.5 displayed marked growth impairment. Tail length and body weight were significantly decreased in the UPM group compared with controls (GD16.5: control n = 72, UPM n = 62; GD21.5: control n = 74, UPM n = 55). (E) Maternal serum hormone profiling at GD21.5 revealed reduced levels of PAPP-A in UPM-exposed dams (G6), with a pronounced ~44% decrease in the sub-group exhibiting reduced litter sizes (G6⁻). Circulating CGα(CGA) levels were also significantly reduced following UPM exposure. Serum samples from 6 control and 9 UPM-treated rats were analyzed. Statistical significance was determined using two-tailed Mann–Whitney test. (F) Epidemiological analysis of n = 994 deliveries from Delhi (high PM2.5 exposure; annual mean 100.9 μg/m³) and Deoghar (PM2.5 < 50 μg/m³) demonstrated a clear exposure–response relationship between ambient PM2.5 levels and adverse birth outcomes. Stratification across exposure gradients (< 50, 60–100, >100 μg/m³) showed progressive declines in neonatal birth weight and increased incidence of preeclampsia, rising from 1.92% at moderate exposure to 6% in the highest category. Ordinary one-way ANOVA indicated significant inverse associations between PM2.5 and neonatal birth weight, with PM2.5 emerging as a significant risk factor for low birth weight and reduced gestational age. All data are presented as mean ± SD. Source data are available online for this figure.

These integrated findings from controlled animal studies and epidemiological data establish that gestational air pollution exposure produces measurable adverse effects on pregnancy outcomes, validating cellular mechanisms identified in our experimental models and confirming significant public health implications of urban air quality on reproductive health.

## Comprehensive proteomic and transcriptomic profiling reveals UPM-induced placental protein dysregulation

To elucidate the molecular networks disrupted by gestational UPM exposure in vivo, we performed proteomic ($n = 3$ control and 3 UPM) and transcriptomic analyses (($n = 2$ control and 3 UPM) of placental tissue from control and UPM-exposed dams at GD16.5 and GD21.5. Principal Component Analysis (PCA) revealed distinct clustering between treatment groups across gestational stages, indicating proteomic reprogramming following UPM exposure (Appendix Figs. S8 and S9). Differential protein expression analysis at GD21.5 (padj<0.05, |Log2FC| ≥0.5) identified 587 differentially expressed proteins, with 287 upregulated and 300 downregulated proteins (Fig. 5A; Appendix Fig. S10). We found that AGPS was the most downregulated (Log2Fold −4.2, $p < 2.2 \times 10^{-6}$) and EPHX2 was among the top upregulated (Log2Fold 2.66, $p < 0.01$) proteins in the GD16.5 UPM exposed groups (Appendix Fig. S8C). Loss of AGPS causes a deficiency of alkylglycerone phosphate synthase, which is crucial for plasmalogen synthesis, cell membrane and myelin formation. AGPS deficiency leads to shortened long bones and intellectual disability (Liegel et al, 2014). EPHX2 in the placenta metabolizes epoxides, which are formed during xenobiotic metabolism, indicating the presence of xenobiotics at the maternal-fetal interface (Gautheron and Jéru, 2020). At GD21.5, proteins associated with DNA damage response, cell death, and DNA repair were upregulated, indicating UPM exposure's genotoxic effects (Fig. 5B–D; Appendix S11). Transcriptomic analysis revealed different profiles between control and UPM groups, shown in PCA plot and heatmap (Fig. 5E). Gene Set Enrichment Analysis (GSEA) showed enrichment of pathways related to inflammatory responses, cell death, and coagulation, confirming particulate matter's harmful effects on placental function.

Hematoxylin and Eosin staining (H&E) (Appendix Fig. S12), Iron staining (Appendix Fig. S13) and TUNEL assays on GD21.5 placental tissue showed marked structural differences with increased iron staining areas marking cell death regions, and features of lipofuscin. The TUNEL assay showed increased TUNEL-positive cells (apoptotic) in the UPM group placenta (Fig. 5F), confirming the findings of proteomic and transcriptomic data.

## Adverse effects of gestational UPM exposure on postnatal neurodevelopment of pups

To assess the long-term consequences of maternal UPM exposure, we performed neurobehavioral evaluations in the offspring of control and UPM-exposed pregnancies following standard weaning. Behavioral phenotyping included assessments of motor coordination (rotarod test), anxiety-like behavior (elevated plus maze, EPM), and spontaneous locomotor and exploratory activity (open-field test). Litter analysis revealed a reduction in the number of viable male pups in the UPM group compared to controls (Control: 6.9 ± 1.4 vs. UPM: 2.2 ± 2.8 males/litter, $p = 0.12$; Fig. 6A), indicating a sex-linked susceptibility to in utero particulate exposure. Elevated plus-maze analysis revealed altered risk-assessment behavior. UPM-exposed pups spent a significantly greater proportion of time in open arms (Control: 37.92 vs. UPM: 49.2, $p = 0.015$) and made fewer closed-arm entries, indicating disrupted anxiety regulation and impaired threat processing. In the rotarod test, pups from UPM-exposed dams exhibited markedly reduced latency to fall (Control: 102.85 ± 7.7 s vs. UPM: 94.96 ± 9.2 s, $p = 0.010$; Fig. 6B), suggesting impaired neuromuscular coordination. In the open-field test, UPM offspring exhibited a pronounced increase in rearing frequency (Control: 16.2 ± 2.6 vs. UPM: 19.6 ± 3.6, $p < 0.000023$) and ~40% more defecation count (Control: 1.05 ± 0.8 vs. UPM: 1.76 ± 1.3, $p = 0.033$), alongside enhanced i.e., 38.7% corner-seeking behavior (Control: 12.7 ± 10.3 s vs. UPM: 20.8 ± 7.7 s, $p = 0.003$), reflecting heightened anxiety and stress responsiveness. Collectively, these findings demonstrate that gestational UPM exposure leads to persistent postnatal deficits in motor coordination, cognitive processing, and emotional regulation, consistent with neurobehavioral disruption and sex-specific vulnerability induced by in utero particulate exposure.

## Urban particulate matter-associated metals transfer across the placenta, impacting offspring inflammatory cascades and developmental outcomes

To verify the causal relationship between UPM exposure and developmental impacts, we performed elemental analysis of the offspring's serum to quantify metallic bioaccumulation following gestational exposure. Inductively coupled plasma mass spectrometry (ICP-MS) revealed elevated concentrations of neurotoxic and developmentally toxic metals in pups serum from UPM-exposed dams, including increases in arsenic, lead, cadmium, zinc, vanadium, nickel, and antimony compared to controls (Fig. 6C). The presence of cadmium exclusively, absent in the control samples, provides evidence of exposure-specific metal accumulation, similar to our cell line studies. These results demonstrate the placental transfer of UPM-associated metallic contaminants from maternal to fetal circulation, confirming that neurodevelopmental deficits may result from direct fetal exposure to toxic metals. Cytokine profiling revealed pro-inflammatory skewing in pups of UPM-exposed animals (Fig. 6D), with elevation of inflammatory cascades, including Th1/Th17-associated cytokines (IL-2, IL-17A, IL-17F, IFN-γ), innate immune activators (IL-1β, TNF), and allergic response mediators (IL-3, IL-5, IL-9, IL-31, IL-33). Chemokine network analysis showed enhanced expression of monocyte/macrophage attractants (CCL4), eosinophil chemoattractant (CCL11), Th2 cell recruiters (CCL17), and neutrophil mobilizers (CXCL1, CXCL11), indicating immune activation. Essential immunosuppressive and tissue repair factors, including FGF21, IL-4, and PDCD1, were reduced, reflecting impaired immune resolution. These results demonstrate efficient placental transfer of UPM-associated metallic contaminants from maternal to fetal circulation, confirming that observed neurodevelopmental deficits possibly result in UPM-exposed offspring, through chronic immune-mediated tissue damage and metabolic disruption.

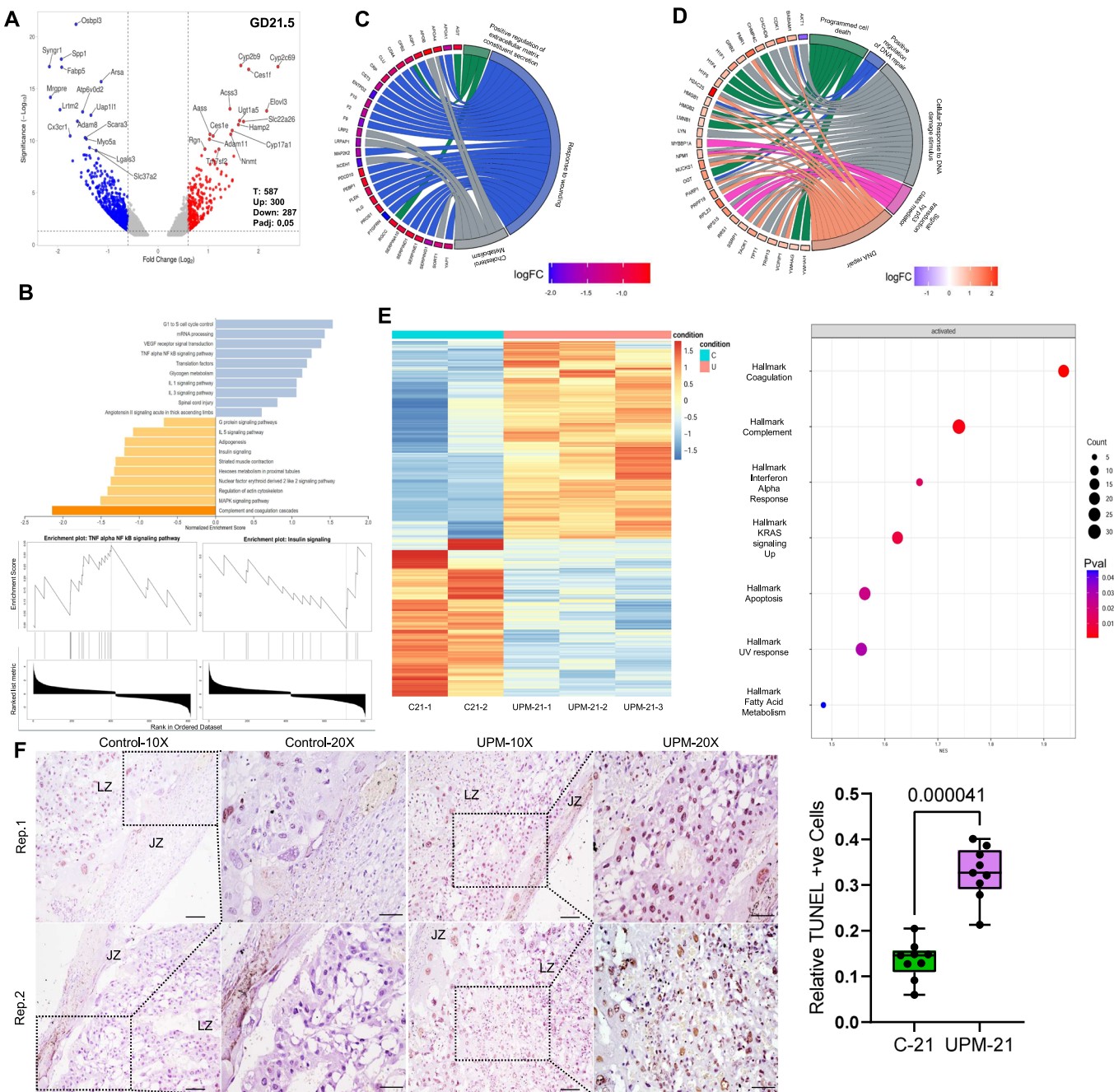

## 16S Ribosomal RNA metagenomic analysis reveals UPM-induced maternal gut dysbiosis and xenobiotic pathway activation

16S rRNA gene sequencing of maternal fecal samples collected at gestational day 21.5 demonstrated significant microbiome community restructuring in UPM-exposed dams relative to unexposed controls (Appendix Figs. S14 and S15). Taxonomic profiling revealed marked enrichment of Prevotellaceae_Alloprevotella species within the UPM treatment group, with this bacterial expansion driving overall community composition changes as evidenced by distinct clustering patterns in principal component

analysis (Appendix Fig. S15). The dominance of Alloprevotella, represents a concerning shift given this genus's established associations with inflammatory activation and intestinal barrier dysfunction. In the context of pregnancy, where maternal immune tolerance and metabolic homeostasis are critical for placental-fetal health, Alloprevotella-driven dysbiosis may disrupt immunological balance and compromise gestational outcomes through systemic inflammatory cascades.

Functional metagenomic analysis revealed profound metabolic pathway alterations reflecting microbial adaptation to environmental contaminant exposure (Fig. 6E; Appendix S15). Significant enrichment of xenobiotic biodegradation pathways including

◀

**Figure 5. Proteomic and transcriptomic profiling reveal key functional pathways disrupted in placentas from UPM-exposed dams.**

(A) Quantitative proteomic profiling of placental tissue at GD21.5 ($n = 3$ per group) identified extensive UPM-induced alterations. Volcano plot analysis revealed 300 upregulated and 287 downregulated differentially expressed proteins (DEPs) in UPM-exposed placentas compared with controls. (B–D) Pathway enrichment analysis identified predominant activation of pro-inflammatory signaling cascades concurrent with suppression of insulin-mediated metabolic signaling networks following UPM exposure. (B) GSEA demonstrated positive enrichment of TNF-α/NF-κB, IL-1, IL-5, and complement/coagulation pathways, whereas insulin signaling and multiple metabolic pathways were negatively enriched. Representative enrichment plots for TNF-α/NF-κB and insulin signaling are shown. (C) Circos network analysis of downregulated DEPs (upper panel) indicated functional clustering in extracellular matrix remodeling and tissue repair pathways. (D) Upregulated DEPs (lower panel) were enriched in apoptotic signaling and DNA damage response programs, suggesting particulate-induced cellular stress and compromised placental integrity. (E) RNA-seq analysis of GD21.5 placentas revealed a distinct transcriptional profile in UPM-exposed samples relative to controls. Differential expression analysis was performed in R package APEGLM as per the standard bioinformatics pipeline, and genes with |log₂FC| ≥ 1 and $p < 0.05$ considered significant. Hierarchical clustering demonstrated broad transcriptional reprogramming, with significant enrichment of pro-apoptotic regulators, cytokine-mediated inflammatory pathways, and cellular stress response genes, indicating disruption of placental physiological homeostasis. (F) TUNEL was conducted on nine placental tissue sections ($n = 9$) obtained at gestational day 21.5. UPM-exposed placental specimens demonstrated significantly elevated numbers and staining intensity of TUNEL-positive cells compared to control tissue, confirming enhanced apoptotic activity as evidenced by brown diaminobenzidine (DAB) chromogenic detection within nuclear compartments. Representative images (Rep1 and Rep2) at 10× and 20× are shown. Scale bar equals 100 μm for 10x and 50 μm for 20x. LZ: Labyrinth zone; JZ: Junctional Zone. The box represents the interquartile range (25th–75th percentiles), the center line denotes the median, and whiskers extend from the minimum to the maximum values. All individual data points are shown. All data are presented as mean ± SD. Statistical significance was determined with $p < 0.05$, using two-tailed Mann–Whitney test. Source data are available online for this figure.

xylene, fluorobenzoate, ethylbenzene, chlorocyclohexane, toluene, and benzoate metabolism confirmed microbiome-mediated responses to pollutant bioaccumulation within the maternal system. Additionally, enhanced porphyrin biosynthetic pathways, central to heme synthesis and oxidative stress regulation, suggest dysregulated cellular energetics that may exacerbate placental oxidative damage, compromise oxygen transport efficiency, and promote cellular apoptosis. The activation of *Helicobacter pylori* epithelial signaling networks indicates microbiome-mediated modulation of host epithelial responses that may amplify inflammatory processes, while altered ansamycin biosynthesis reflects microbial competition dynamics under environmental stress conditions.

## Mechanistic investigation for identifying the mediator responsible for UPM induced placental and fetal abnormalities

Through multi-platform analysis of UPM-stimulated HTR8/SVneo cells, first-trimester placental explants, and third-trimester rat placental proteomics, dysregulation of insulin-like growth factor signaling networks was observed, particularly in the IGF-binding protein family. Statistical analysis with stringent thresholds identified insulin-like growth factor binding protein 3 (IGFBP3) as significantly downregulated across all experimental groups, suggesting a critical role for IGF pathway disruption in UPM-mediated placental dysfunction. Proteomic data of GD21.5 placenta showed that IGFBP3 was highly abundant in control placentas while no signal was detected in placentas of UPM group (Fig. 7A). Evaluation of IGFBP3 expression in placental tissue of control and UPM-exposed rats at GD21.5 revealed a significant downregulation at both the mRNA level (2.43-fold, $p = 0.003$, $n = 10$; Fig. 7B) and protein level, as confirmed by immunohistochemical analysis (~2-fold, $p = 0.4 \times 10^{-4}$, $n = 9$; Fig. 7C), reaffirming its marked abundance in control placentas. Placental tissue sections demonstrated significantly reduced IGFBP3 protein localization compared to control animals, with quantitative analysis revealing approximately 40% reduction in IGFBP3 immunoreactivity within the junctional zone and labyrinth regions critical for nutrient transport and fetal growth regulation. These findings confirm that inflammatory response due to the UPM can disrupt the expression of IGFBP3 and thus show its adverse effects in affecting placentation

and fetal growth. To elucidate the regulatory mechanisms underlying this pathway disruption, we investigated the relationship between UPM-induced inflammatory responses and IGFBP3 gene and protein expression, focusing on how pro-inflammatory cytokines including TNF-α, IL-1β, and IFN-γ might suppress IGFBP3 expression. HTR8/SVneo cells stimulated with UPM, IL1β or vehicle were used to assess the change in protein expression of IGFBP3. UPM and IL-1β significantly reduced the expression of IGFBP3 (Fig. 7D,E). These findings were further supported by analyses of public datasets on placental SARS-CoV-2 infection, pollutant exposure, and pregnancy complications, including pre-eclampsia (PE) and gestational diabetes mellitus (GDM) (Meng et al, 2012; Ganguly et al, 2024; Kallol et al, 2023; Li et al, 2023), all of which consistently showed a downregulation of IGFBP3 (Fig. 7F; Appendix S16A). The pharmacological rescue of IGFBP3 expression through STAT1 inhibition with fludarabine provides definitive evidence for the mechanistic role of inflammatory signaling in mediating UPM effects (Fig. EV3A). This functional validation strongly supports the computational prediction of STAT1 response elements in the IGFBP3 promoter (Fig. EV3B). This experimental validation corroborates the in silico identification of putative STAT1 binding motifs within the IGFBP3 upstream regulatory region (Fig. EV3B,C). To determine whether reduced IGFBP3 is associated with impaired fetal growth, we assessed IGFBP3 levels in the serum of pups (post-weaning) from control and UPM-exposed dams. Dot blot analysis revealed a marked reduction in IGFBP3 (1.77-fold, $p = 0.0033$) (Fig. 7G; Appendix S16B) in the UPM group compared to controls, suggesting that UPM-mediated downregulation of IGFBP3 may contribute to compromised postnatal health.

## Discussion

Air pollution exposure during pregnancy risks maternal health, placental function, and postnatal health (Aguilera et al, 2023; Mitku et al, 2023). Associations include adverse outcomes like fetal growth restriction, low birth weight, gestational diabetes, hypertension, and preeclampsia (Mitku et al, 2023; Aguilera et al, 2023). David M. Stieb et al, found 1 ppm CO exposure reduced birth weight by 11.4 g, while PM2.5 (10 μg/m³) and PM10 (20 μg/m³) reduced it by 23.4 and 16.8 g, respectively (Stieb et al, 2012). Indoor and outdoor

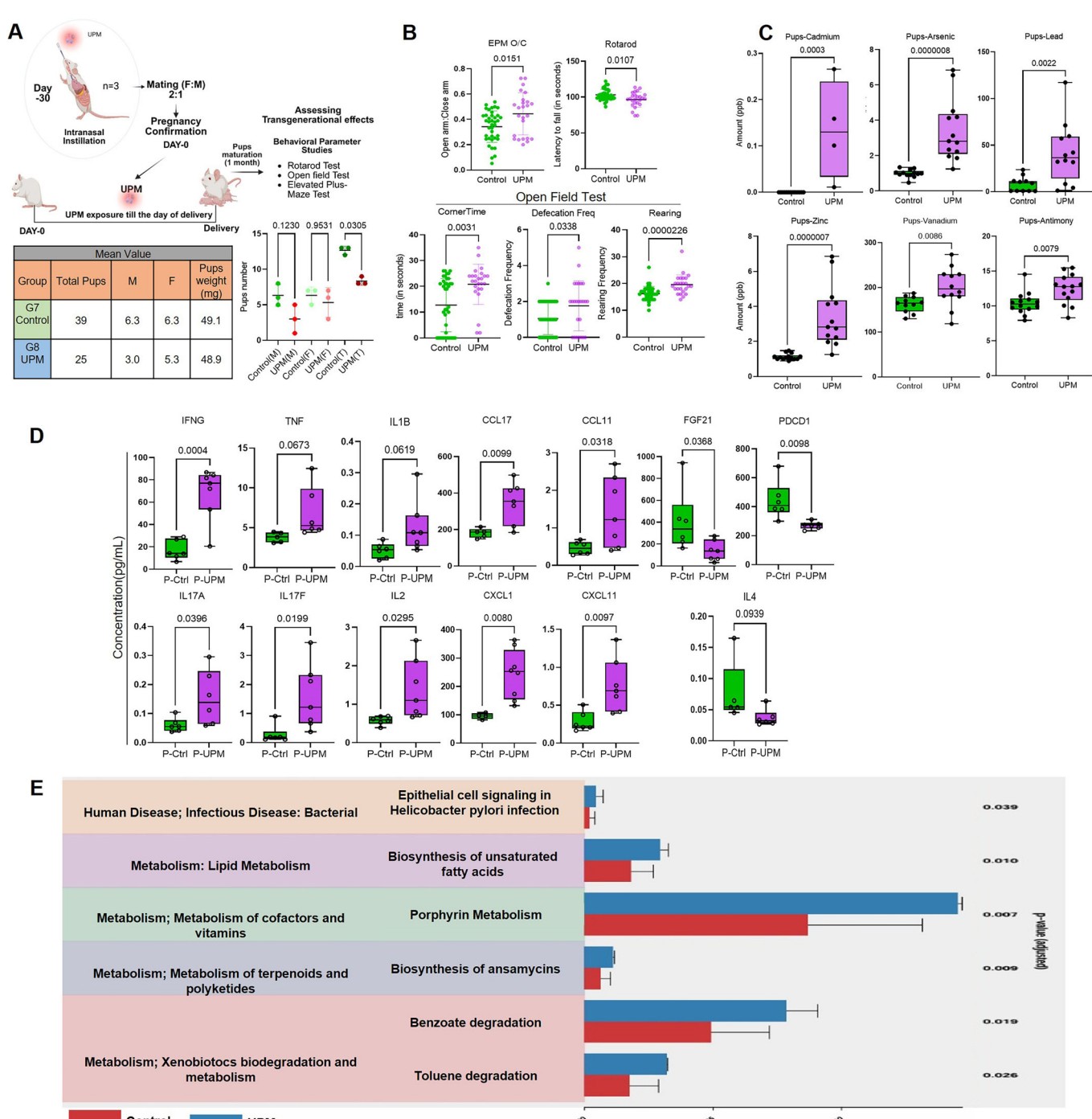

pollutants were linked to LBW incidence (Ghosh et al, 2021; Gong et al, 2022; Li et al, 2020). Air pollution affects birth outcomes through changes in hematocrit, blood viscosity, endothelial function, and inflammation (Baskurt et al, 1990; Risom et al, 2005). Data links air pollution and particulate matter to adverse outcomes, but mechanistic insights into effects on placentation, maternal-fetal immune homeostasis, and toxic metal transfer remain unclear. Our investigation, using cellular models, tissue explants, animal studies, and clinical correlations, reveals a framework where UPM exposure disrupts placental processes such

as trophoblast invasion, syncytialization, endocrine function, and nutrient transport. The mechanisms identified show UPM exposure triggers pathological events like endoplasmic reticulum stress and epigenetic changes, impairing trophoblast functions. The reduction in trophoblast migration, invasion and syncytialization in our assays indicates compromised extravillous and syncytiotrophoblast functions essential for artery remodeling, placental vascularization, barrier formation and endocrine function of the placenta, which are often the hallmark features of pregnancy complications including preeclampsia and intrauterine growth restriction (Azmi et al, 2024;

**Figure 6. Urban particulate matter exposure leads to maternal gut dysbiosis, placental metal transfer, inflammatory responses and postnatal behavioral alterations in offspring.**

(A) Schematic overview of the UPM exposure regimen with accompanying quantitative data on litter outcomes. UPM-exposed dams delivered ~50% fewer male pups than controls, indicating potential sex-specific vulnerability or selective embryonic/fetal loss. M: Male; F:Female; T: Total. (B) Elevated plus maze showed, UPM-exposed pups spent significantly more time in open arms (Control: 37.92 s vs. UPM: 49.2 s, $p = 0.015$), indicating impaired anxiety regulation. Rotarod test showed offspring from UPM-treated dams showed reduced latency to fall (Control: 102.85 ± 7.7 s vs. UPM: 94.96 ± 9.2 s, $p = 0.010$), reflecting neuromotor deficits. Open field test showed UPM-exposed pups displayed heightened anxiety and stress responses, with increased rearing (Control: 16.2 ± 2.6 vs. UPM: 19.6 ± 3.6, $p = 0.000023$), greater defecation (Control: 1.05 ± 0.8 vs. UPM: 1.76 ± 1.3, $p = 0.033$), and prolonged corner-sitting (Control: 12.7 ± 10.3 vs. UPM: 20.8 ± 7.7 s, $p = 0.003$). Offspring numbers: Control ($n = 39$) and UPM ($n = 25$). (C) ICP-MS quantification revealed significantly elevated serum levels of multiple toxic metals in UPM-exposed pups ($n = 13$/group). Notably, cadmium was uniquely detected only in UPM offspring and completely absent in controls, indicating selective cadmium transfer and systemic bioaccumulation. (D) Multiplex analysis ($n = 6$) demonstrated marked upregulation of pro-inflammatory cytokines (IFN-γ, IL-17A, IL-17F, IL-2), chemokines (CCL17, CCL11, CXCL1, CXCL11), and trends toward elevated TNF-α and IL-1β. Anti-inflammatory mediators (FGF21, PDCD1) were significantly reduced, indicating diminished immunoregulatory capacity and potentiation of chronic inflammation. Statistical analysis included serum from six different pups(P) from control and UPM treated dams ($n = 6$ biological replicates). The box represents the interquartile range (25th–75th percentiles), the center line denotes the median, and whiskers extend from the minimum to the maximum values. All individual data points are shown. (E) 16S rRNA sequencing revealed significant gut dysbiosis in UPM-exposed dams, with enrichment of *Prevotellaceae_Alloprevotella* and upregulation of xenobiotic degradation and porphyrin biosynthesis pathways. These microbial shifts suggest pollutant-driven metabolic adaptation but also potential amplification of oxidative stress and inflammatory burden affecting maternal–fetal health. All data are presented as mean ± SD. Statistical significance was determined with $p < 0.05$, using two-tailed Mann-Whitney test. Source data are available online for this figure.

Liang et al, 2023). Studies have shown that perturbed expression of fusion mediating proteins such as Syncytin-1, GCM1 and βhCG leads to the altered placental morphology, embryonic lethality and are also linked to IUGR and preeclampsia (PE) (Jeyarajah et al, 2022; Vargas et al, 2011). Gestational exposure of UPM in rodents revealed a significant decline in litter size (12.5% at gestational day (GD) 16.5 to 25% at GD 21.5), placental weight and diameter with features of compromised cellular architecture. There were also features of fetal resorption which clinically resembles either implantation failure or miscarriage. These experimental observations justify and align with existing epidemiological findings, where PM10 exposure from biomass fuel increased spontaneous abortion risk (aOR: 3.12, 95% CI: 1.07, 4.17) (Mukherjee et al, 2015), and analysis of 255,668 pregnant women in Beijing (2009–2017) showed associations of PM2.5, SO₂, O₃, and CO with missed abortion in the first trimester in 6.8% women, with increasing ORs for each pollutant (Zhang et al, 2019). Placenta is the key source of nutrients, gases and growth factors for the developing fetus and any compromise in its structure would inversely affect the fetal growth (Regnault et al, 2002), which was evident from the reduced fetal weight and growth (34% decrease by term) along with corresponding alterations in maternal serum pregnancy hormones (40–55% reduction in β-hCG and 35–48% decrease in PAPP-A). These findings are particularly significant given that both β-hCG and PAPP-A serve as established clinical biomarkers for placental insufficiency and pregnancy complications (Kosinski et al, 2023; Ghasemi-Tehrani et al, 2017). A South Asian study showed PM2.5 exposure (mean 56 µg/m³) contributed to 7.1% yearly pregnancy losses above 40 µg/m³ levels (Xue et al, 2021). Meta-analysis further showed PM2.5 exposure associated with hypertensive disorders (OR = 1.52) and preeclampsia (OR = 1.31) (Sun et al, 2020), with similar findings by Yu H et al (Yu et al, 2020) and others. These results parallel epidemiological data from India, China, the United States, and Europe, which report dose-dependent associations between PM2.5 exposure and 20–30 g birth weight reductions per 10 µg/m³ increase, along with higher risks of hypertensive disorders, preeclampsia, and spontaneous abortion (Stieb et al, 2012; Bekkar et al, 2020; Capobussi et al, 2016). Thus, our observed reductions in placental growth, fetal weight, litter size, and pregnancy hormone levels not only establish mechanistic insights into placental insufficiency and fetal growth restriction but also

substantiate and provide biological justification for the adverse pregnancy outcomes reported.

Our metagenomic analysis reveals an additional layer of complexity in UPM-mediated effects through disruption of maternal gut microbiome composition and function. The significant enrichment of Prevotellaceae_Alloprevotella and activation of xenobiotic degradation pathways demonstrates that environmental pollutants fundamentally alter microbial ecosystem dynamics in ways that may contribute to systemic inflammation and metabolic dysfunction. This microbiome-mediated pathway represents an important area for future therapeutic intervention and highlights the complex, multi-system effects of environmental pollutant exposure (Lear et al, 2021; Singh et al, 2022). The link between microbiota dysbiosis and systemic inflammation has been well-established in the scientific literature. Studies have shown that alterations in gut microbiome composition can lead to increased intestinal permeability, allowing bacterial products to enter the bloodstream and trigger systemic inflammatory responses (Lama Tamang et al, 2023; Di Vincenzo et al, 2024). In the context of environmental pollutants, recent studies have found that exposure to particulate matter and other air pollutants can induce changes in gut microbiota composition, leading to increased inflammation and metabolic disturbances (Rio et al, 2024; Cao et al, 2023). These findings underscore the complex interplay between environmental exposures, microbiome dysbiosis, and systemic inflammation, supporting the observations made in our metagenomic analysis.

Our transcriptomic analysis in HTR8/SVneo, explants and rat placental tissue revealed that UPM exposure triggers stress responses and robust activation of pro-inflammatory cascades. The significant upregulation of cytokines including IL-1β, IFN-γ, TNF-α, and multiple chemokines (CCR/CCL family members) demonstrates that particulate matter exposure creates a sustained inflammatory environment that is detrimental to normal placental function. This is particularly important as the immune system is an indispensable part of the feto-maternal unit, participating in multiple facets of decidualization and placenta formation by shaping trophoblast invasion, syncytialization, vascular remodeling, and differentiation (Hussain et al, 2022; Joo et al, 2024). Cytokines such as IL-10 produced by dendritic cells are crucial for generating regulatory T-cells, and activated dendritic cells support angiogenesis at the fetal-maternal interface by producing VEGF-A, FGF-2,

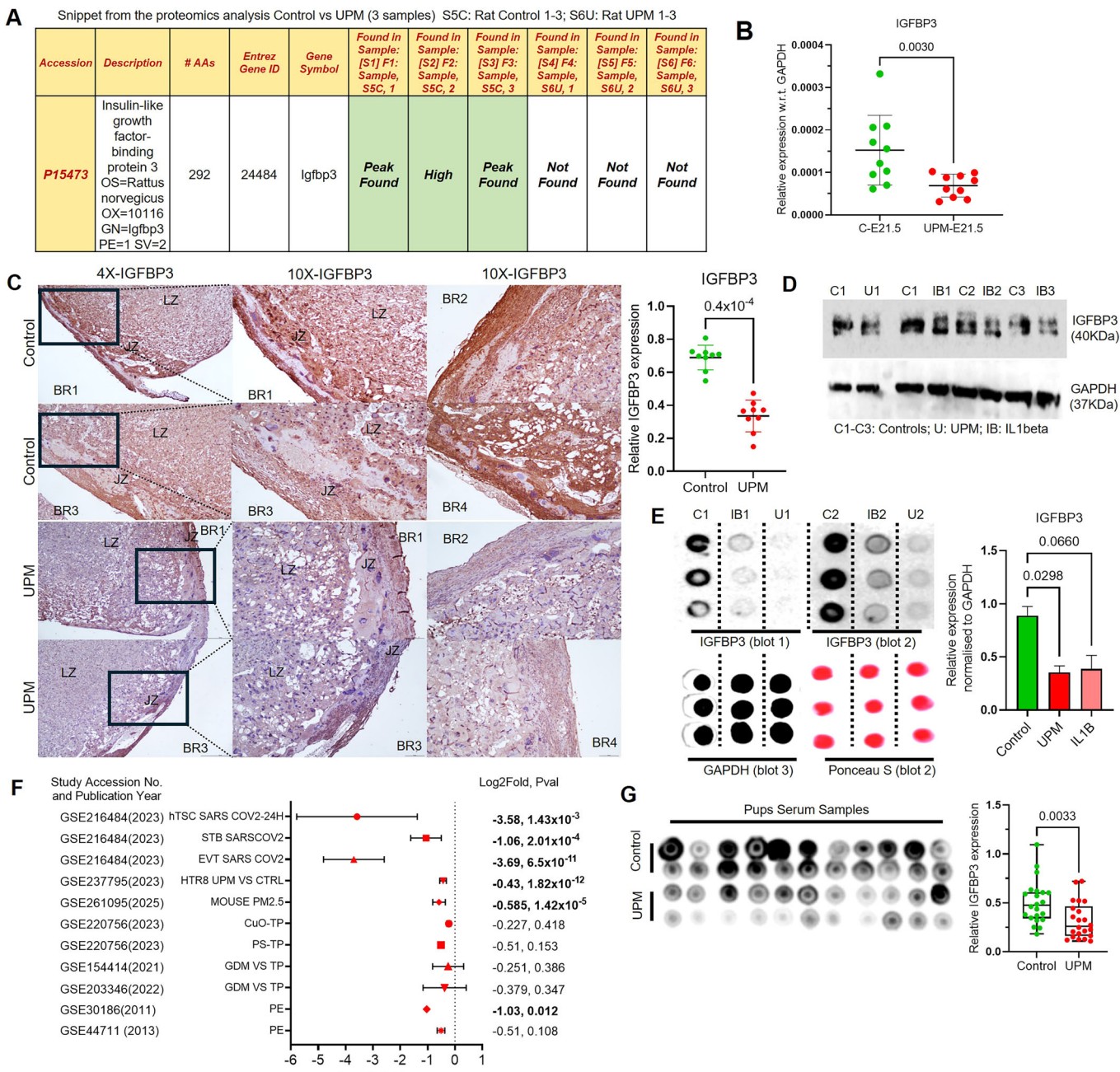

**A** Snippet from the proteomics analysis Control vs UPM (3 samples) S5C: Rat Control 1-3; S6U: Rat UPM 1-3

| Accession | Description | # AAs | Entrez Gene ID | Gene Symbol | Found in Sample: [S1] F1: Sample, S5C, 1 | Found in Sample: [S2] F2: Sample, S5C, 2 | Found in Sample: [S3] F3: Sample, S5C, 3 | Found in Sample: [S4] F4: Sample, S6U, 1 | Found in Sample: [S5] F5: Sample, S6U, 2 | Found in Sample: [S6] F6: Sample, S6U, 3 |
|---|---|---|---|---|---|---|---|---|---|---|
| P15473 | Insulin-like growth factor-binding protein 3 OS=Rattus norvegicus OX=10116 GN=Igfbp3 PE=1 SV=2 | 292 | 24484 | Igfbp3 | Peak Found | High | Peak Found | Not Found | Not Found | Not Found |

**B** IGFBP3

**C** 4X-IGFBP3 · 10X-IGFBP3 · 10X-IGFBP3 · IGFBP3

**D** C1 U1 C1 IB1 C2 IB2 C3 IB3 — IGFBP3 (40KDa), GAPDH (37KDa) · C1–C3: Controls; U: UPM; IB: IL1beta

**E** C1 IB1 U1 C2 IB2 U2 — IGFBP3 (blot 1), IGFBP3 (blot 2), GAPDH (blot 3), Ponceau S (blot 2) · IGFBP3

**F** Study Accession No. and Publication Year — Log2Fold, Pval

| | Log2Fold, Pval |
|---|---|
| GSE216484(2023) hTSC SARS COV2-24H | **-3.58, 1.43x10⁻³** |
| GSE216484(2023) STB SARSCOV2 | **-1.06, 2.01x10⁻⁴** |
| GSE216484(2023) EVT SARS COV2 | **-3.69, 6.5x10⁻¹¹** |
| GSE237795(2023) HTR8 UPM VS CTRL | **-0.43, 1.82x10⁻¹²** |
| GSE261095(2025) MOUSE PM2.5 | **-0.585, 1.42x10⁻⁵** |
| GSE220756(2023) CuO-TP | -0.227, 0.418 |
| GSE220756(2023) PS-TP | -0.51, 0.153 |
| GSE154414(2021) GDM VS TP | -0.251, 0.386 |
| GSE203346(2022) GDM VS TP | -0.379, 0.347 |
| GSE30186(2011) PE | **-1.03, 0.012** |
| GSE44711 (2013) PE | -0.51, 0.108 |

**G** Pups Serum Samples — IGFBP3

Endothelin-I, and CXCL8 (Lv et al, 2025; Wei et al, 2021). Disruption of immune balance or inflammatory cascades due to pollutants potentially affects this homeostasis (Glencross et al, 2020; Croft et al, 2021). Multiple studies have shown that particulate matter exposure induces inflammation via oxidative stress-triggered pathways such as ERK, JNK-MAPK, and NFkB, contributing to disease onset in respiratory and cardiovascular systems (Xu et al, 2020; Nääv et al, 2020). Recent evidence underscores that PM2.5 and PAH-rich air pollution exerts strong inflammatory and oxidative effects across maternal, placental, and fetal compartments (Craig et al, 2024; Agarwal et al, 2018; Dai et al, 2023). Elevated maternal PAH metabolites such as 1-hydroxypyrene and benzo[a]pyrene correlate with higher

placental TNF-α ($p < 0.05$) and reduced IL-6, indicating a TNF-α-dominant milieu (Ferguson et al, 2017). Personal exposure monitoring similarly shows that each unit increase in PAH-bound PM2.5 corresponds to higher umbilical cord TNF-α, IL-8, and TGF-β ($p \approx 0.01$–0.02), demonstrating fetal inflammatory coupling to maternal exposure (Zhang et al, 2025). Prenatal NO₂ exposure associates with increased autophagy and senescence markers (Beclin-1, β ≈ 0.13, 95% CI 0.04–0.23) and elevated IL-1β, IL-8, and MMP-9 clusters (global $p \approx 0.002$), linking pollutant stress to remodeling at birth (Gorlanova et al, 2025). Supporting this, the MADRES cohort found PM2.5-induced enrichment of oxidative stress and inflammatory lipid pathways inversely correlated with birth weight (Chen et al, 2025). Experimental

◀ **Figure 7. Mechanistic investigation reveals that UPM-induced placental and fetal abnormalities are mediated by decreased IGFBP3 expression.**

(A) Proteomic profiling (LC-MS/MS) of placental tissue at GD21.5 ($n = 3$ per group) detected robust IGFBP3 protein signals in all control samples, whereas IGFBP3 was completely absent in UPM-exposed placentas, indicating strong UPM-associated suppression of IGFBP3 at term. (B) qPCR analysis of GD21.5 placental tissue revealed significant downregulation of IGFBP3 transcripts in UPM-exposed dams relative to controls ($n = 10$; $p = 0.003$). Relative fold change ($2^{-\Delta Ct}$) was normalized to GAPDH (housekeeping gene). (C) Immunohistochemistry demonstrated markedly reduced IGFBP3 protein staining in UPM-treated placentas ($n = 9$), with the most prominent loss observed in the LZ. ImageJ-based quantification confirmed significant reduction in IGFBP3-positive staining intensity ($p = 0.4 \times 10^{-4}$), corroborating proteomic and mRNA results. LZ: Labyrinth zone; JZ: Junctional zone. Scale bar equals 100 μm. (D) In HTR8/SVneo trophoblast cells, Western and dot blot analyses showed substantial IGFBP3 reduction following UPM (U) or IL-1β (IB) treatment compared with untreated controls (C) (C1–C3, U1–U3, IB1–IB3), indicating cytokine- and pollutant-driven repression of IGFBP3 protein. (E) Dot blot assays were performed in technical triplicates and repeated across three independent experiments. Blots 1 and 2 represent biological replicates for IGFBP3 detection, while Blot 3 (GAPDH) and Ponceau S staining verified equal protein loading. Densitometric quantification (ImageJ) demonstrated significant IGFBP3 suppression in UPM and IL-1β-treated cells. (F) Meta-analysis of publicly available RNA-seq datasets revealed consistent downregulation of IGFBP3 across diverse pathological or environmental contexts—including PE, GDM, infection, and nanoparticle exposure. Forest plot includes $\log_2$ fold change values, $P$-values, and GSE accession numbers. CuO: copper oxide nanoparticles; PS: polystyrene nanoplastics; TP: term placenta. Comparisons with $p > 0.05$ were considered non-significant. (G) Dot blot analysis of circulating IGFBP3 levels in offspring serum demonstrated significantly reduced systemic IGFBP3 in UPM-group pups compared with controls (Control (C1–C22); UPM (U1–U22): $n = 22$), indicating persistence of IGFBP3 suppression into the postnatal period and potential long-term effects on growth factor signaling. The box represents the interquartile range (25th–75th percentiles), the center line denotes the median, and whiskers extend from the minimum to the maximum values. All individual data points are shown. All data are shown as mean ± SD. Statistical significance was assessed using two-tailed Mann–Whitney test. Source data are available online for this figure.

models parallel these findings: PM2.5 elevates IL-6, IL-8, and TNF-α ($p < 0.05$) and induces ROS-dependent PI3K/AKT activation, driving NF-κB/JNK signaling, oxidative injury, and insulin resistance (Lu et al, 2025). Similar kinase-mediated inflammation (MKK4–JNK–AP1 axis) in keratinocytes shows that MKK4 inhibition suppresses IL-6, COX-2, and MMP-1 (Liao et al, 2025; Rundblad et al, 2025). Together, these studies reveal a conserved pattern wherein PM2.5/PAHs trigger ROS accumulation, cytokine dysregulation, and stress-kinase activation that disrupt metabolic homeostasis, prime immune pathways, and compromise placental–fetal development, providing mechanistic context for pollutant-induced placental inflammation in our study (Rundblad et al, 2025; Liao et al, 2025; Yun and Kim, 2025; Lu et al, 2025). Particulate matter can reach the placenta and fetal side, impacting pregnancy outcomes. Inflammation, a known contributor to disorders like preeclampsia, FGR, gestational hypertension, and gestational diabetes, has been epidemiologically linked to PM2.5, PM10 exposure and adverse outcomes (Singh et al, 2024, 2025; Jung et al, 2024). Diesel Exhaust Particle (DEP) exposure further alters inflammatory biomarkers postnatally, affecting offspring growth (Tsai et al, 2021). The identification of similar inflammatory signatures across our cellular models, tissue explants, and animal studies provides strong evidence that particulate matter exposure induces chronic inflammation, disrupting trophoblast function and immune homeostasis at the maternal-fetal interface, thereby establishing inflammation as a key pathway underlying UPM-mediated placental dysfunction and pregnancy complications.

One of the most significant findings of this investigation is the identification of IGFBP3 as a critical target of UPM-induced dysfunction. The consistent downregulation of IGFBP3 across multiple experimental platforms, coupled with our mechanistic validation of IL-1β/STAT1-mediated suppression, establishes a novel pathway linking environmental exposure to growth factor signaling disruption. The partial restoration of IGFBP3 expression following STAT1 inhibition suggests that inflammatory signaling may contribute to the regulation of the IGF axis under UPM-induced stress conditions. This observation is compatible with the presence of predicted STAT1-responsive elements within the IGFBP3 promoter and supports a model in which cytokine-driven STAT1 activation participates in IGFBP3 repression. While

these findings do not establish direct causality, they provide convergent experimental and in silico evidence that STAT1-dependent transcriptional mechanisms may be involved in mediating inflammation-associated modulation of IGFBP3. Collectively, these data demonstrate that UPM exposure possibly activates STAT1, potentially acting synergistically with NF-κB-mediated pathways or independently, resulting in transcriptional silencing of IGFBP3. The phylogenetic conservation of these cis-regulatory elements across mammalian species indicates that this inflammation-mediated transcriptional control mechanism constitutes an evolutionarily preserved biological pathway connecting environmental stress responses, possibly leading to perturbations in placental gene expression and embryo development programs, ultimately manifesting as compromised maternal-fetal health outcomes. While these in silico findings suggest a transcriptional control by STAT1/NFκB, their functional validation at the chromatin and transcriptional level remains to be established and is currently beyond the scope of this work.

While direct evidence linking PM2.5, PM10 exposure to IGFBP3 repression is limited, convergent findings from PAH toxicology, endocrine disruptor studies, and inflammation research provide strong mechanistic plausibility (Talia et al, 2021). PM2.5 is a major carrier of PAHs and transition metals that activate the AhR–CYP1A1 axis, generating excessive ROS, mitochondrial dysfunction, and oxidative DNA damage (Dai et al, 2023). The resultant oxidative stress and redox-sensitive pathways (NF-κB, PI3K/AKT, MAPK) elevate pro-inflammatory cytokines including IL-1β, TNF-α, and IFN-γ, potent activators of the JAK/STAT1 cascade (Mahmoud et al, 2021; Gambini and Stromsnes, 2022). PM2.5 exposure also upregulates the redox-sensitive transcription factor KLF9, which enhances CYP1A1 expression and contributes to oxidative stress, mitochondrial dysfunction, and impaired placental morphogenesis (Yuan et al, 2019; Li et al, 2023). Cytokine-driven STAT1 activation is known to repress IGFBP3 transcription, as seen in chronic inflammatory states such as cystic fibrosis and cachexia, where elevated IL-1β, IL-6, and TNF-α correlate with reduced IGF/IGFBP3 bioavailability (Martín et al, 2021; Taylor et al, 1997). Recent trophoblast and placental studies show that PAH and PM2.5-induced CYP1A1 activation drives mitochondrial apoptosis and cytokine upregulation, linking

pollutant metabolism to inflammatory stress (Li et al, 2023; Nääv et al, 2020). Endocrine disruptors—including dioxins, bisphenol A, and phthalates—likewise lower IGFBP3 through ROS-dependent STAT1/NF-κB activation and promoter methylation, providing epidemiological support for pollutant-driven IGFBP3 loss (Huang et al, 2020; Wu et al, 2017; Boas et al, 2010; Su et al, 2015). IGFBP3 functions not only downstream of cytokine stress but also as a regulator of mitochondrial stability (Street et al, 2003; Lee et al, 2011). IGFBP3 depletion results in swollen, fragmented mitochondria, increased fission, reduced MFN2, elevated BNIP3L/NIX, enhanced mitophagy, suppressed mTOR activity, and reduced mtDNA—hallmarks of oxidative injury (Stuard et al, 2022). Thus, reduced IGFBP3 in PM2.5-exposed trophoblasts likely intensifies mitochondrial destabilization initiated by PAH-activated CYP1A1. Together, these findings support a model in which PM2.5-derived PAHs activate the KLF9–CYP1A1 oxidative axis, while ROS and cytokine signaling suppress IGFBP3, removing a key mitochondrial stabilizer. This synergy explains the mitochondrial dysfunction, trophoblast impairment, and placental insufficiency associated with particulate pollution exposure, positioning IGFBP3 repression as a central mechanism contributing to disrupted IGF signaling and fetal growth restriction.

Given that IGFBP3 serves as the primary regulator of IGF-1 bioavailability and is essential for fetal growth, this pathway disruption provides a mechanistic explanation for the intrauterine growth restriction observed in our animal models and clinical correlations (Kaur et al, 2021; Nawathe et al, 2016) IGFBP3, the predominant IGF-binding protein in maternal serum at term, plays a critical role in regulating IGF bioavailability by binding 70–80% of circulating IGF-I and IGF-II, thereby prolonging their half-life and maintaining a stable reservoir for growth regulation (Varma Shrivastav et al, 2020). In the placenta, IGFBP3 helps modulate the transfer of IGFs to the fetus, influencing nutrient delivery and growth-promoting signaling (Forbes and Westwood, 2008; Carter et al, 2006).

A reduction in IGFBP3, whether at the placental level during gestation or in the postnatal circulation of offspring, can disrupt this regulatory balance, increasing IGF turnover and limiting its bioavailability for receptor-mediated signaling in target tissue. This deficit can impair fetal growth trajectories, as evidenced in small-for-gestational-age (SGA) infants, where lower IGFBP3 expression has been reported (Renes et al, 2019). Furthermore, IGFBP3 possesses IGF-independent actions, including the regulation of cell proliferation and induction of apoptosis, which are essential for normal placental development and tissue remodeling (Varma Shrivastav et al, 2020). Its downregulation may therefore exacerbate pregnancy complications by impairing placental structure and function, contributing to adverse outcomes such as fetal growth restriction and compromised postnatal development.

The demonstration of persistent neurobehavioral deficits in offspring from UPM-exposed dams represents a critical finding with significant public health implications. Motor coordination deficits, altered anxiety responses, and compromised cognitive function observed in postnatal behavioral testing establish that gestational UPM exposure produces lasting effects extending beyond the prenatal period. Elevated metallic constituents (arsenic, lead, cadmium, zinc) detected in offspring serum provide direct evidence for transplacental transfer of particulate-bound contaminants, with cadmium exclusively present in exposed animals

serving as a definitive biomarker confirming efficient maternal-fetal transfer of toxic metals. Exposure to these metallic ions is associated with aberrant inflammatory responses, as supported by our high-throughput cytokine profiling showing significant inflammation in pups from UPM-exposed dams, suggesting that transplacental metal transfer drives systemic immune activation. Epidemiological studies have similarly linked maternal immune activation to neurological disorders in offspring (Loayza et al, 2023; Han et al, 2021). Choi et al demonstrated that increased IL-17a-producing T-cells in pregnant mice elevated fetal IL-17a, disrupting cortical development and inducing autism-like behaviors (Choi et al, 2016), while Hsiao et al showed that maternal IL-6 administration altered endocrine signaling and fetal neurobehavioral outcomes (Hsiao and Patterson, 2011). Together, our findings align with existing evidence demonstrating that gestational pollutant exposure, via transplacental transfer and immune activation, results in adverse neurodevelopmental outcomes in offspring.

The correlation of our experimental findings with epidemiological data from New Delhi, India provides critical validation of clinical relevance. The consistent increase in preeclampsia incidence (2% to 6%) and reduction in birth weight across PM2.5 exposure establishes clear connections between air quality and pregnancy outcomes. Human pregnancy is defined by prolonged gestation, advanced fetal organ maturation at birth, and a haemochorial placenta specialized for endocrine, metabolic, and immune regulation; however, no single animal model fully recapitulates all these features (Andersen et al, 2018). While large mammals and non-human primates offer closer anatomical similarity, their ethical, financial, and logistical constraints limit mechanistic and multi-omic interrogation (Carter, 2020). Rodent models (rat and mouse) provide a haemochorial placental interface, conserved trophoblast signaling pathways, robust genetic tractability, and experimental scalability, making them optimal discovery platforms for identifying evolutionarily conserved molecular mechanisms (Suarez et al, 2024; Soares et al, 2012; Ramdin et al, 2023; Aguilera et al, 2022). Rodents, like humans, possess a hemochorial placenta with invasive trophoblast lineages and share conserved IGF/IGFBP-regulated growth, oxidative stress susceptibility, and AhR/CYP1A1 responses to PAHs (Mason et al, 2011; Shynlova et al, 2007; Choi et al, 2012; Lew et al, 2011; Karube et al, 2024). These conserved features, together with supporting evidence from our human trophoblast and placental explant experiments, strengthen the translational relevance of our observations. Nonetheless, differences in villous architecture, depth of invasion, and the temporal dynamics of placental maturation mean that our findings should not be interpreted as direct quantitative predictors of human outcomes. Rather, they reveal conserved qualitative mechanisms through which PM2.5 perturbs placental development.

These findings have immediate implications for clinical practice, suggesting that air quality monitoring should be integrated into prenatal care protocols, particularly in urban environments with high pollution levels. The mechanistic pathways identified in our study integrate closely with the broader epidemiological literature showing that maternal PM2.5 exposure impairs fetal growth, increases preeclampsia risk, and contributes to preterm birth. Large multinational meta- analyses consistently report pregnancy-averaged PM2.5 exposures associated with reductions in birth

weight. Our findings provide molecular insight into these associations. Specifically, PM2.5 induced activation of CYP1A1 generates oxidative and ER stress culminating in IGFBP3 repression, thereby impairing trophoblast function and nutrient support. Our results also align with epidemiologic studies linking PM2.5 to hypertensive disorders of pregnancy such as preeclampsia, where elevated TNF-α, IL-6, and IL-1β, STAT1/NF-κB activation, and angiogenic imbalance (reduced PlGF/VEGF with increased sFlt-1) are prominent features—molecular signatures mirrored in our model. The inflammation-driven ER-stress and apoptosis we observed similarly provide plausible biological links to the increased preterm birth risk reported in human cohorts. This work identifies ER stress as a central pathway of PM2.5 toxicity and positions IGFBP3 and CYP1A1 as interconnected oxidative–inflammatory–endocrine nodes governing placental vulnerability. These pathways represent potential biomarkers and therapeutic targets for high-risk pregnancies. Studies on PAHs, heavy metals, and traffic pollutants confirm that diverse PM2.5 components converge on similar oxidative/ER-stress mechanisms, unifying epidemiological and molecular evidence that PM2.5 disrupts placental development through integrated cellular injury pathways.

## Limitations and future directions

While this study provides robust evidence for UPM-mediated placental dysfunction, several limitations must be acknowledged. Reliance on animal models, although valuable for mechanistic insights, may not fully recapitulate human placental development and function. Moreover, real-world pollution exposure involves multiple pollutants and temporal variations, which are simplified in controlled studies. Future research should validate these findings in human populations through prospective cohorts with detailed exposure assessments and biomarker analyses. Developing intervention trials to test protective strategies identified here is a critical next step for clinical translation. Further investigation into microbiome-mediated effects, epigenetic mechanisms, and the long-term consequences of gestational UPM exposure on offspring health through lifespan-focused longitudinal studies are also needed to fully understand transgenerational impacts. Importantly, an in-depth investigation into the role of IGFBP3 as a central mediator of UPM-induced placental and fetal alterations is warranted. Deciphering how inflammatory signaling regulates IGFBP3 repression, and how this axis contributes to adverse pregnancy outcomes and postnatal developmental trajectories, will be essential to establish causal mechanisms and identify therapeutic opportunities.

## Methods

### Reagents and tools table

| Reagent/Resource | Reference or Source | Identifier or Catalog Number |
| --- | --- | --- |
| **Experimental models** | | |
| HTR8/SVneo cells | ATCC | CRL-3271 |
| BeWo cells | ATCC | CCL-98 |
| Rattus Norvegicus | AIIMS, New Delhi | |

| Reagent/Resource | Reference or Source | Identifier or Catalog Number |
| --- | --- | --- |
| **Antibodies** | | |
| MMP2 | Abclonal | A19080 |
| MMP9 | Abclonal | A0289 |
| TIMP1 | Abclonal | A1389 |
| TIMP2 | Abclonal | A1558 |
| XBP1s | Abclonal | A1731 |
| NRF2 | Abcam | ab62352 |
| IRE1A | Cell Signaling | #3294 |
| Bip | Abclonal | A11366 |
| NFkB | Abcam | ab32536 |
| Syncytin1 | Immunotag | ITAB91578 |
| CHOP | Cell Signaling | #2895 |
| IGFBP3 (Human) | Cell Signaling | #25864 |
| IGFBP3 (Rat) | Elabscience | E-AB-91410 |
| GAPDH | Affinity | AF7021 |
| Anti-Rabbit-HRP | Cell Signaling | 7074S |
| Anti-Mouse-HRP | Cell Signaling | 7076 |
| Anti-Rabbit-Alexa Fluor 594 | Invitrogen | A11029 |
| Anti-Mouse-Alexa Fluor 488 | Invitrogen | A11012 |
| **Oligonucleotides** | | |
| MMP2-F | AGGATGGCAAGTACGGCTTC | |
| MMP2-R | CTTCTTGTCGCGGTCGTAGT | |
| MMP9-F | GAGCTGACTCGACGGTGATG | |
| MMP9-R | AACTGTATCCTTGGTCCGGG | |
| TIMP1-F | CATTGCTGGAAAACTGCAGGA | |
| TIMP1-R | GCAGTTTGCAGGGGATGGAT | |
| TIMP2-F | GGCTGCGAGTGCAAGATCAC | |
| TIMP2-R | TCGAGAAACTCCTGCTTGGG | |
| UPA-F | CCCAGGAAATGGGACAGGG | |
| UPA-R | ACAGTTCGCCTGTTCGTATCT | |
| PAI-F | CATCCTGGAACTGCCCTACC | |
| PAI-R | AGGGAGAACTTGGGCAGAAC | |
| IL1B-F | AGCTACGAATCTCCGACCAC | |
| IL1B-R | CGTTATCCCATGTGTCGAAGAA | |
| TNFA-F | CCTCTCTCTAATCAGCCCTCTG | |
| TNFA-R | GAGGACCTGGGAGTAGATGAG | |
| IL6-F | ACTCACCTCTTCAGAACGAATTG | |
| IL6-R | CCATCTTTGGAAGGTTCAGGTTG | |
| IL10-F | TCAAGGCGCATGTGAACTCC | |
| IL10-R | GATGTCAAACTCACTCATGGCT | |
| IL4-F | ATGGGTCTCACCTCCCAACT | |
| IL4-R | GATGTCTGTTACGGTCAACTCG | |
| IL17-F | AGATTACTACAACCGATCCACCT | |

| Reagent/Resource | Reference or Source | Identifier or Catalog Number |
|---|---|---|
| IL17-R | GGGGACAGAGTTCATGTGGTA | |
| TGFB-F | CTAATGGTGGAAACCCACAACG | |
| TGFB-R | TATCGCCAGGAATTGTTGCTG | |
| MCP1-F | CAGCCAGATGCAATCAATGCC | |
| MCP1-R | TGGAATCCTGAACCCACTTCT | |
| IL23-F | TCCTGTCTTGCATTGCACTAAG | |
| IL23-R | CATCCTGGTGAGTTTGGGATTC | |
| IFNA-F | ACTCATACACCAGGTCACGC | |
| IFNA-R | CAGTGTAAAGGTGCACATGACG | |
| IL23-F | CTCAGGGACAACAGTCAGTTC | |
| IL23-R | ACAGGGCTATCAGGGAGCA | |
| IFNG-F | TGACAGAAAAATAATGCAGAGCCA | |
| IFNG-R | TGGACATTCAAGTCAGTTACCGAA | |
| SYN1-F | CTGTTGGACTTACTTCACCCAAA | |
| SYN1-R | GGTACGGAGGGTTTCATGTAGT | |
| HCGB-F | CTACTGCCCCACCATGACCC | |
| HCGB-R | GCAGAGTGCACATTGACAGC | |
| GCM1-F | TTCTCCAAGAGTTATGGTCTGGG | |
| GCM1-R | CCACGCTTGTAGATCGCCA | |
| SYN2-F | AGCCCCTATTTGTGTTATGGC | |
| SYN2-R | GGAATTGGTTGTGGGTGTATGT | |
| MFSD2A-F | CCATTGATGAGGAGAGGCGG | |
| MFSD2A-R | CCTTCTGTGGCCTTCTGCAT | |
| DYSFERLIN-F | AAGAACAGCGTGAACCCTGTA | |
| DYSFERLIN-R | CCTCTCGGAGTGGGACCTT | |
| GAPDH-F | ACGGATTTGGTCGTATTGGG | |
| GAPDH-R | CGCTCCTGGAAGATGGTGAT | |
| Rat-IGFBP3-F | GCGGGAGACAGAATATGGTCC | |
| Rat-IGFBP3-R | AGAACCCCTTCTTGTCACAGT | |
| Rat-GAPDH-F | CACCATCTTCCAGGAGCGAG | |
| Rat-GAPDH-R | TCACAAACATGGGGGCATCA | |
| **Chemicals, Enzymes and Other reagents** | | |
| Urban Particulate Matter (UPM) | Sigma-Aldrich (NIST SRM 1648a) | SRM 1648a |
| CellTracker Green CMFDA Dye | Molecular Probes, Invitrogen | C7025 |
| CellTracker Orange CMTMR Dye | Molecular Probes, Invitrogen | C2927 |
| Calcein-AM | Invitrogen, Thermo Fisher Scientific | C1430 |
| DAPI | Invitrogen, Thermo Fisher Scientific | D3571 |
| Propidium Iodide | Sigma-Aldrich | P4170 |
| Methylcellulose | Sigma-Aldrich | M0512 |
| Bovine Collagen Type I | ThermoFisher Scientific Inc. | A1064401 |
| Matrigel solution | Becton Dickinson and Company | 356234 |
| Masson´s Trichrome Stain Kit | G-Biosciences | BAQ085 |

| Reagent/Resource | Reference or Source | Identifier or Catalog Number |
|---|---|---|
| CellROX® Oxidative Stress Reagent | Life Technologies (Thermo Fisher Scientific) | C10444 |
| Forskolin | Sigma-Aldrich | F6886 |
| RPMI-1640 | Himedia | AL028A |
| DMEM:F12 | Himedia | AL140S |
| Fetal Bovine Serum (FBS) | GibcoTM, ThermoFisher Scientific Inc. | 10270106 |
| Penicillin–Streptomycin Solution | Himedia Laboratories Private Ltd. | A002A |
| Triton X-100 | Thermo Fisher Scientific | AAA16046AE |
| Coomassie Brilliant Blue R-250 | Biorad | 1610436 |
| Gelatin | Sigma, St. Louis | G-8150 |
| Trypsin–EDTA Solution | Himedia Laboratories Private Ltd. | TCL 007 |
| PVDF Membranes | Biorad | 1620177 |
| NC Membranes | Biorad | 1620112 |
| Protease inhibitor cocktail | G Biosciences | 786-108 |
| Phosphatase inhibitor cocktail | G Biosciences | 786-870 |
| Protein molecular weight standards | GeneDireX | PM008-0500 |
| **Software** | | |
| GraphPad Prism 9.0 | GraphPad Software, Inc | |
| ImageJ | https://imagej.nih.gov/ij/index.html | |
| BioRender | https://www.biorender.com/ | |
| Axion BioSystems | | |
| FastQC (v0.11.9) | Andrews, 2010 | |
| Trimmomatic (v0.39) | Bolger et al, 2014 | |
| STAR (v2.7.4a) | Dobin et al, 2013 | |
| HTSeq-count (v0.11.2) | Anders et al, 2015 | |
| DESeq2 (R package) | Love et al, 2014 | |
| pheatmap | Kolde, 2025 | |
| EnhancedVolcano | Blighe et al, 2023 | |
| ClusterProfiler | Yu et al, 2012 | |
| enrichplot | Yu, 2019 | |
| Proteome Discoverer v2.5 | Thermo Fisher Scientific, USA | |
| QIIME2 | Bolyen et al, 2019 | |
| SILVA Database (v138) | Quast et al, 2013 | |
| Tax4Fun | Aßhauer et al, 2015 | |
| ggPicrust2 | Douglas et al, 2020 | |
| ALDEx2 (R package) | Bioconductor, USA | |
| **Others** | | |
| EZcountTM WST-1 Cell Assay Kit | Himedia Laboratories Private Ltd. | CCK032 |
| Total RNA isolation kit | Promega Corporation | Z6011 |

| Reagent/Resource | Reference or Source | Identifier or Catalog Number |
|---|---|---|
| Nanodrop | Thermo Fisher Scientific, Inc | |
| Verso cDNA synthesis Kit | Thermo Fisher Scientific, Inc. | AB1453A |
| DyNAmo Flash SYBR Green | Thermo Fisher Scientific, Inc. | F415S |
| Bicinchoninic Acid (BCA) assay | G Biosciences | 786-570 |
| Enhanced chemiluminescence (ECL) detection kit | Thermo Fisher Scientific, Inc. | NCI4106 |
| hCG-beta ELISA Kit | Elabscience | E-EL-H0175 |
| Rat-PAPPA ELISA Kit | Krishgen | KLR0526 |
| Rat-CGalpha ELISA Kit | Krishgen | KLR3633 |
| EpiQuik Total Histone Extraction Kit | Epigentek | OP-0006-100 |
| EpiQuik Histone H3 Modification Multiplex Assay Kit | Epigentek | P-3100-96 |
| TUNEL In Situ Apoptosis Kit | Elabscience | E-CK-A331 |
| TrueSeq Stranded total RNA Kit | Illumina | 20020597 |
| Qubit 4.0 fluorometer | Thermofisher | Q33238 |
| Tapestation 4150 | Agilent | |
| D1000 screentapes | Agilent | 5067-5582 |
| Illumina NovaSeq 6000 V1.5 | | |

## Cell culture and treatment

Human extravillous trophoblast cells HTR8/SVneo (obtained from ATCC) and human choriocarcinoma cells BeWo (obtained from ATCC) were used in this study. While HTR8/SVneo was used to study trophoblast invasion, fusogenic BeWo cells were used to study syncytialization. The cells were grown in RPMI-1640 medium (Hyclone, USA), supplemented with 10% Fetal bovine serum (FBS-Gibco) and 1% Penicillin/Streptomycin (Gibco, USA). They were maintained at 5% $CO_2$ and 37 °C. Urban Particulate matter (UPM-NIST 1648a) was purchased from sigma Aldrich and used to prepare a solution of 50 mg/mL in 9:1 (1x-PBS:DMSO). The final working concentration of the UPM used was 50 μg/mL. The working solution was briefly sonicated for 5 min before using it for stimulation studies. For control/unstimulated cells equal volume of 1X PBS:DMSO was added. EZcount™ water-soluble tetrazolium dye (WST-1) Cell Assay kit (CCK032-Himedia, India) was used to observe the effect on cell viability following the manufacturer's instructions.

## EZcount™ water-soluble tetrazolium dye (WST-1) cell assay

EZcount™ WST-1 Cell Assay Kit (Catalog: CCK032, Himedia Laboratories Private Ltd., Mumbai, India) was used to study the

effect of UPM doses on the viability of cells by following the manufacturers protocol. To assess the effect of UPM on cells and to identify the LD50 dose 10,000 cells/well were plated in 96-well plate in triplicates. After 12 h of plating cells were exposed to different concentration of UPM ranging from 10 μg/mL to 400 μg/mL for 48 h in fresh complete media. After 48 h 10 μL activated WST solution was added to each well and incubated at 37 °C for 2 h. Color development was monitored and recorded using a colorimeter at 450 nm and 625 nm. Results were analyzed by comparison with standard curve.

## Cell cycle assessment

$2 \times 10^5$ HTR8/SVneo and BeWo cells were grown in each well of a 6-well culture plate. After 12 h of plating cells were treated with 50 μg/mL of UPM for 48 h. Thereafter cells were collected by trypsinization and were fixed in 70% sterile ethanol for 10 min at 4 °C. Cells were then centrifuged at $1000 \times g$ for 10 min at 4 °C followed by washing three times with sterile 1X-PBS. Cell pellet was resuspended in 500 μL of 1X-PBS containing 20 μg/mL of RNaseA at 37 °C for 30 min. 8 μL of 10 μg/mL of PI solution was added to each tube and mixed well. Cell suspension was transferred to flow tubes and cells were incubated on ice for 20 min in dark. A total of 10,000 events/runs were analyzed using flow cytometer.

## CellRox assay

Cells were plated in 12-well plates. After treatment of cells, as mentioned above, 5 μM CellRox (Life Technologies C10444) was added to each well and incubated at 37 °C for 30 min. The cells were then washed 3 times with 1X PBS at room temperature and analyzed by flow cytometry using BD FACSCanto™.

## Inductively coupled plasma–mass spectrometry (ICP-MS)

UPM exposed cells were washed thrice with serum free RPMI-1640 and 1 ml from each wash was stored separately for downstream analysis. Cells were harvested by trypsinization in a 15 mL tube and pelleted at $1000 \times g$ at room temperature. Pelleted cells were resuspended in 1 mL serum free RPMI-1640 and transferred to a 1.5 mL micro-centrifuge tube. Again, cells were pelleted, and the supernatant was discarded. Cell pellet was dried at 40 °C on a hot plate. Subsequently, cell lysis is performed to release intracellular components, including metallic ions. The protein concentration of the resulting lysate is determined through BCA assay and is adjusted to a consistent level across samples to facilitate normalization of ICP-MS results. The samples undergo digestion using hot block digestion, and post-digestion, they were cooled and diluted with deionized water for analysis.

For ICP-MS calibration, a series of calibration standards were prepared using a multi-element standard, covering the expected concentration range of metallic ions of interest. Internal standards were included in each calibration standard to correct for variations during analysis. The sample analysis phase involves introducing the prepared samples and calibration standards into the ICP-MS instrument. Calibration standards are run to establish a calibration curve. UPM-exposed samples were then analyzed, ensuring the stability and calibration of the instrument. During analysis, metallic

ions of interest were monitored, and their concentrations was quantified based on the established calibration curve. This comprehensive procedure ensured accurate and reliable assessment of metallic ion content in UPM-exposed samples.

## Transmission and scanning transmission electron microscopy (TEM/STEM) analysis

HTR8/SVneo cells ($1 \times 10^6$) exposed to UPM for 48 h, along with control cell in a 25 cm² flask, were processed for ultrastructural and elemental analysis. Cell pellets were fixed in Karnovsky's fixative (2.5% glutaraldehyde + 2% paraformaldehyde in 0.1 M phosphate buffer, pH ~7.4) at 4 °C, rinsed twice in phosphate buffer, and post-fixed with 1% osmium tetroxide for 1 h. Following fixation, samples were dehydrated in graded acetone series, infiltrated, and embedded in Araldite CY212 resin, which was polymerized at 60 °C. Ultrathin sections (60–70 nm) were cut using an ultra-microtome, mounted on copper grids, and stained with uranyl acetate and lead citrate.

For ultrastructural assessment, sections were examined on a Talos 200S transmission electron microscope (Thermo Scientific, Inc.) at the All India Institute of Medical Sciences. Digital images were captured at fixed magnifications, and quantitative analysis of subcellular organelles was performed.

To identify UPM-associated toxic metals within the cells, Scanning Transmission Electron Microscopy (STEM) coupled with Energy-Dispersive X-ray Spectroscopy (EDS) was employed. STEM-EDS mapping of ultrathin sections enabled elemental detection and localization of cadmium (Cd), zinc (Zn), antimony (Sb), and lead (Pb) within intracellular compartments. These analyses provided direct evidence of UPM-derived metal accumulation in HTR8/SVneo cells and correlated with observed ultra-structural perturbations.

## Placental explant culture

All procedures involving human participants were conducted in accordance with the ethical standards of the institutional ethics committee and conformed to the principles set out in the WMA Declaration of Helsinki and the Department of Health and Human Services Belmont Report. Human placental tissues were processed as described previously. Early gestation placentas (8–12 weeks, $n = 6$) from medically terminated pregnancies and term placentas (37–40 weeks, $n = 6$) from uncomplicated deliveries were used. Placentae were placed maternal side up, and small cotyledon blocks ($3 \times 3 \times 3$ mm) were dissected and washed with ice-cold PBS. Samples were cultured in collagen-I coated 35 mm petri plates (Corning) with DMEM:F12 containing 10% FBS and 1% penicillin-streptomycin at 37 °C with 5% $CO_2$. Explants were treated with 50 µg/mL UPM or vehicle for 48 h, followed by collection in TriZol for RNA isolation.

## Quantitative polymerase chain reaction (RT-qPCR)

$2 \times 10^5$ HTR8/SVneo cells plated in a 6-well plate were stimulated with 50 µg/mL UPM for 48 h. $2 \times 10^5$ BeWo cells were either stimulated with 10 µM Forskolin alone or Forskolin with the 50 µg/mL UPM for 48 h. RNA from cells was extracted following the kit's instructions (Promega RNA Tissue/Cell Miniprep System). Verso

cDNA synthesis kit (AB1453A, Thermo Fisher Scientific, USA) was used for cDNA preparation, and DyNAmo Flash SYBR Green qPCR kit (F415S, Thermo Fisher Scientific, USA) was used for gene expression analysis. Sequences for PCR primers are provided in Reagents and Tools Table.

## Animal exposure and behavioral studies

The study was conducted after getting permission from the institutional animal ethics committee. Institutional ethical clear-ance was obtained vide letter #F.02/IAEC/CAF/2021-2022, dated: Oct. 10, 2022. Wistar rats (Rattus Norvegicus) were used, including 46 females (4–6 weeks old) and 21 males (6–8 weeks old). Animals were acclimatized for at least five days, housed three per cage, and identified by tail marking and cage cards. After acclimatization, female rats were randomized into eight groups (G1–G8). Half received 35 µL normal saline via intranasal instillation for 4 weeks, while the other half received 35 µL UPM. The UPM dose was calculated based on human equivalent exposure using the method of Lehman and Fitzhugh (1954) (Lehman, 1954; Vermillion et al, 2018), considering an average tidal volume of 2.5 mL and a respiratory rate of 100/minute for Wistar rats, resulting in a total daily air intake of 0.36 m³/day. In New Delhi, pregnancy exposure to PM2.5 ranges from 26.7–77.3 µg/m³, the rat equivalent exposure was calculated as 50 µg/m³ × 0.36 m³/day × 20 days × 100 (uncertainty factor) = 36,000 µg, replicating human pregnancy exposure levels. Females were mated with males at a 2:1 ratio, and pregnancy was confirmed by daily vaginal smear examination for sperm presence. Pregnant rats were then assigned to groups: G1, G3, G5, and G7 received saline, while G2, G4, G6, and G8 received UPM. Each group had 6 rats. Treatments were administered from gestational day (GD) 1–7 (G1, G2), GD 1–16 (G3, G4), and GD 1–21 (G5, G6, G7, G8). Animals in G1–G6 underwent caesarean section after exposure, while G7 and G8 delivered naturally. Endpoints included daily body weight and clinical observations, serum collection at termination, pup count, weight, tail length measurement, and placental weight and morphology assessment (half fixed in formalin, half snap frozen). Fetal growth parameters were recorded, and after caesarean, animals and pups were euthanized by $CO_2$ asphyxiation. Behavioral tests were performed on weaned pups (3–4 weeks old) from G7 and G8, including Rotarod (motor coordination and balance), Open Field Test (exploratory behavior and anxiety), and Elevated Plus-Maze Test (anxiety-related behavior), with data interpreted for motor, locomotor, exploratory, and anxiety-related outcomes.

## Human demographic study

Institutional ethics approval was taken to use the delivery records of pregnant women at AIIMS, New Delhi and Deoghar. Institu-tional ethical clearance was obtained vide letter #IECPG-499/30.06.2022, dated: July 01, 2022. For this study, we investigated the effect of PM2.5 exposure on low birth weight and preeclampsia (PE) by comparing pregnancy outcomes between an exposed site, AIIMS New Delhi, where PM2.5 levels range from 60 to over 100 µg/m³, and a control site, Deoghar, with PM2.5 levels below 50 µg/m³. Pregnancy records were collected retrospectively from both sites for deliveries occurring between 2022 and 2024. Inclusion criteria encompassed pregnant women with singleton pregnancies

who delivered at gestational age ≥30 weeks and had complete medical records including antenatal history, delivery details, and neonatal birth weight. Only women residing within the respective study sites were included to ensure exposure classification. Exclusion criteria comprised multiple pregnancies, known major fetal congenital anomalies or chromosomal abnormalities, pre-existing chronic conditions such as hypertension, pre-gestational diabetes mellitus, chronic kidney disease, or autoimmune disorders, incomplete medical records lacking essential variables, and women who underwent elective deliveries. These criteria ensured a robust data quality and minimized confounding in assessing the relationship between PM2.5 exposure, low birth weight, and PE. A total of 994 records were included for analysis after applying the predefined inclusion and exclusion criteria.

## High throughput cytokine profiling

We utilized the Olink Target 48 Cytokine Panel to analyze 45 analytes in serum samples, adhering to the manufacturer's protocol. Briefly, each assay run processed forty samples on a $48 \times 48$ integrated fluidic circuit (IFC). Pups 1 µl serum samples were incubated with antibodies tagged with DNA sequences for 18 h. After hybridization, DNA tags were extended and preamplified using a Bio-Rad T100 Thermocycler. Preamplified samples and primers were loaded onto a primed IFC, and qPCR was performed for 40 cycles on an Olink Signature Q-100 instrument. Each run included internal controls and calibrators to ensure accuracy. Data analysis and quality control were conducted using Olink NPX Signature software, which converted NPX values to protein concentrations (pg/mL) using standard curves.

## Transcriptomic analysis

Raw FASTQ reads were quality-checked using FastQC (v0.11.9) (Andrews, 2010) and summarized with MultiQC, followed by adapter and low-quality base trimming using Trimmomatic (v0.39) (Bolger et al, 2014). For mRNA-seq, reads were aligned to the human reference genome (GRCh38.p13) with STAR (v2.7.4a) (Dobin et al, 2013), and gene-level counts were obtained using HTSeq-count (v0.11.2) (Anders et al, 2015) with Ensembl GENCODE 42 annotations. Differential expression analysis was performed in R using DESeq2 (Love et al, 2014), with *lfcShrink* ("apeglm") applied to moderate effect sizes (Scull et al, 2024), and genes with $|\log_2 FC| \geq 1$ and $p < 0.05$ considered significant. Visualizations were generated with *pheatmap* (Kolde, 2025) and *EnhancedVolcano* (Blighe et al, 2023). To assess functional relevance, Gene Set Enrichment Analysis (GSEA) was conducted using *ClusterProfiler* (Yu et al, 2012) with genes ranked by $\log_2 FC$ against KEGG and HALLMARK gene sets (MSigDB; size 3–500), considering results with nominal $p < 0.05$ as significant. Normalized Enrichment Scores (NES) indicated activation (positive) or repression (negative), and findings were visualized with *enrichplot* (Yu, 2019) using dot plots and enrichment plots highlighted leading-edge genes and pathway interactions.

## Proteomic analysis

Placental tissue collected from 3 control and 3 UPM treated rats were homogenized in Tris-Cl buffer with SDS and inhibitors, and proteins (25 µg) were reduced, alkylated, and digested using FASP with trypsin. Peptides were purified on C18 cartridges and analyzed on an Easy-nLC 1000 coupled to an Orbitrap Exploris 240 mass spectrometer. MS acquisition used a data-dependent top-20 method, and spectra were searched in Proteome Discoverer v2.5 against UniProt with 1% FDR filtering.

Processed data were $\log_2$ transformed, filtered to retain proteins with ≥70% valid replicates, imputed for missing values, and quantile normalized. Differential abundance between control and treated groups was assessed by Student's t-test, with proteins at $p < 0.05$ considered significant. Z-scaling enabled visualization by heatmaps and correlation plots. Gene set enrichment analysis was performed using WebGestalt (Wang et al, 2017).

## Microbiome analysis

Paired-end FASTQ files from 16S rRNA sequencing were quality-checked (FastQC/MultiQC), trimmed with Trimmomatic, and processed in QIIME2 using DADA2 for denoising, error correction, merging, and chimera removal, yielding high-resolution ASVs. Taxonomy was assigned with vsearch against SILVA 138 (99% similarity) using a pre-trained Naive Bayes classifier. The resulting OTU table was analyzed in MicrobiomeAnalyst after filtering low-abundance and low-variance features, followed by Total Sum Scaling. Alpha diversity (Chao1, Shannon, Simpson) and beta diversity (Bray–Curtis, PCoA, PERMANOVA) were computed. Functional potential was inferred with Tax4Fun and KEGG pathways annotated via ggPicrust2. Differential abundance of pathways related to UPM exposure was assessed with ALDEx2.

## Western blotting

Western blotting was performed as described previously (Singh et al, 2025). HTR8/SVneo and BeWo cells $(0.8 \times 10^6)$ were plated in 25 cm² flasks and treated with 50 µg/mL UPM for 48 h; BeWo cells also received 10 µM Forskolin alone or with UPM. After washing with ice-cold PBS, cells were lysed in RIPA buffer with inhibitors, and lysates centrifuged at 15,000 r.c.f. for 10 min at 4 °C. Protein content was measured by BCA assay, and 30 µg was separated by 10% SDS-PAGE and transferred to 0.2 µm PVDF membranes. Blots were probed with primary and HRP-conjugated secondary antibodies (Reagents and Tools Table & Appendix Table 1). Bands were visualized by ECL and quantified with ImageJ, normalized to GAPDH.

## Dot-blot

For dot blot analysis, serum samples were diluted in phosphate-buffered saline (PBS) to a uniform protein concentration, and equal volumes (1 µL) were spotted directly onto a nitrocellulose membrane. The membrane was air-dried for 30 min, briefly rinsed in TBS, and blocked with 5% non-fat dry milk in Tris-buffered saline containing 0.1% Tween-20 (TBST) for 1 h at room temperature to prevent non-specific binding. It was then incubated for 2.5 h at room temperature (while rocking) with the primary antibody against IGFBP3, diluted in blocking buffer. Following three 10 min washes with TBST, the membrane was incubated with horseradish peroxidase (HRP)-conjugated secondary antibody for 1 h at room temperature. After additional 3 times TBST washes,

signal detection was performed using enhanced chemiluminescence (ECL) substrate, and spot intensities were quantified by densitometric analysis using ImageJ software.

## Gelatin zymography

Substrate gel gelatin zymography was used to assess the bioactivity of cells secreted matrix metalloproteinases (MMPs)–2/9 in the culture media of HTR-8/SVneo cells exposed to UPM. Cell culture media was centrifuged at $10,000 \times g$ for 10 min at 4 °C to get rid of the floating debris and cells. Supernatant was transferred to a new tube and stored at −80 °C until further use. Protein quantification was done using BCA method. 15 μg of proteins were separated using an 8% polyacrylamide gel containing 0.1% gelatin. The gel was then washed with 2.5% Triton X-100 solution for 1 h to renature proteins and incubated in activation buffer (150 mM NaCl, 10 mM $CaCl_2$, 50 mM Tris, and 0.025% sodium azide) at 37 °C with gentle shaking overnight so that separated proteases can show their activity by degrading the gelatin in gel. Gelatinolytic activity was assessed by staining the gel with 0.1% Coomassie brilliant blue R250 for 30 min. The gel was destained using 10% acetic acid solution and visualized.

## Enzyme-linked immunosorbent assay (ELISA)

Equal amounts of protein were used for all ELISA assays. β-hCG ELISA was performed on culture supernatants collected from UPM-treated BeWo cells following the manufacturer's instructions. ELISAs for Pregnancy-Associated Plasma Protein A (PAPPA) and chorionic gonadotropin alpha subunit (CGα) were carried out on rat serum samples according to the respective manufacturer protocols.

## Wound healing assay for migration

$2.5 \times 10^5$ HTR8/SVneo cells were cultured in 6-well plates until they were 80–90% confluent. A scratch was made through the cell monolayer and washed thrice to remove any floating debris or unattached cells. A 50 μg/mL UPM containing media was added to the cells, and gap filling was recorded using time-lapsed video imaging. Images of the wound were taken at regular intervals using an axion Cyto-smart microscope at 4× magnification till the wound was closed. The images were automatically uploaded to the Axion Portal, where the Scratch Analysis Module determined the area closure.

## Matrigel invasion assay

The surface of the insert plate was coated with 50 μL Matrigel solution (400 μg/ml) (356234; Becton Dickinson and Company, NJ, USA, 1:5 in RPMI-1640) and kept at 37 °C for 1 h. HTR8/SVneo cells were plated on the coated insert plate. RPMI-1640 media containing 50 μg/mL UPM was added to the reservoir well. After 24 h of incubation, cells were fixed by 70% ethanol solution. Cells from the inside of inserts were removed using wet cotton swabs. The inserts were placed in Masson's Trichrome solution to stain the cells with insert membranes. The membrane was cut out precisely and mounted in a slide to capture the images using an upright microscope.

## CellTracker staining (fusion assay)

$1 \times 10^6$ BeWo cells were stained with 2.5 μM CellTracker Green CMFDA or CellTracker Orange CMTMR (Molecular Probes, Invitrogen) for 30 min at 37 °C in the dark. Excess dye was removed by centrifugation and washing with 1X PBS, and pellets were resuspended in pre-warmed RPMI-1640 with 10% FBS. The two populations (40,000 cells each) were counted and co-plated in 12-well plates, then treated with FK or FK + UPM. After 24 h, images were captured, and fusion efficiency was analyzed using ImageJ.

## Terminal deoxynucleotidyl transferase-mediated dUTP nick-end labeling (TUNEL) assay

TUNEL assay was performed with Rat placental tissue section using Elabscience® TUNEL In Situ Apoptosis Kit (HRP-DAB Method) (E-CK-A331) following manufacturer protocol. Results were quantified using imageJ.

## Iron detection in paraffin-embedded tissue sections by Perls' Prussian blue staining

Paraffin-embedded tissue sections (4–5 μm thick) were first deparaffinised in xylene twice for 5 min each, followed by rehydration through a graded ethanol series: 100% ethanol twice for 3 min each, 95% ethanol once for 3 min, and 70% ethanol once for 3 min, then rinsed in distilled water for 2 min. For iron detection, a working Perls' Prussian blue solution was freshly prepared by mixing equal volumes of 2% potassium ferrocyanide and 2% hydrochloric acid immediately before use. Sections were immersed in the staining solution for 25 min at room temperature to allow ferric iron to react and form ferric ferrocyanide (blue precipitate). Following staining, slides were rinsed in distilled water three times for 2 min each. For counterstaining, sections were incubated in 0.1% Nuclear Fast Red solution for 4 min, then rinsed in running tap water for 5 min. Sections were dehydrated through 95% ethanol once for 2 min and 100% ethanol twice for 2 min each, cleared in xylene twice for 5 min each, and mounted with a resinous mounting medium under a coverslip. Under light microscopy, ferric iron deposits appeared blue, while nuclei and tissue architecture counterstained with Nuclear Fast Red appeared pink to red.

## Immunocytochemistry (ICC)

To immunolocalize specific antigens, cells were seeded in poly-L-lysine (Merk, Germany) coated coverslips in 35 mm Petri plates. Following 48 h of UPM treatment, 4% paraformaldehyde was used to fix the cells, and ICC was performed as per the standard protocol. Details of antibodies are given in Reagents and Tools Table & Appendix Table 1. Images were taken using a confocal microscope.

## 3D Spheroid formation assay

HTR8/SVneo cells were used to generate spheroids. Control and UPM-treated cells (48 h) were trypsinised, counted, and 5000 cells were mixed with 0.16% methylcellulose and 10 μg/mL bovine

collagen type-I in 10% RPMI-1640, then added to U-bottom 96-well plates. Spheroid formation was monitored over five days with regular imaging. After 5 days, spheroids were used for confocal imaging or migration assays. For imaging, spheroids were stained with 1 µM Calcein, 20 µg/mL propidium iodide, and 100 nM DAPI for 15 min before confocal microscopy. For migration assays, spheroids were plated on 0.1% gelatin-coated 96-well plates and cultured for 48 h, then fixed with 70% ice-cold ethanol for 10 min and stained with 0.1% crystal violet for 2 min before imaging by phase-contrast microscopy.

## Effect of UPM on epigenetics modifications

Total histone fractions were isolated from control and UPM treated cells after 48 h, using EpiQuik Total Histone Extraction Kit (Epigentek, Farmingdale, USA) following manufacturer protocol. Protein quantification for normalization was done using the BCA method. Isolated histone fraction was used for Histone-3 modification ELISA using EpiQuik Histone H3 Modification Multiplex Assay Kit (Colorimetric) following manufacturer's instructions. This ELISA kit is optimized for assessing 21-different histone modifications in one plate. Added Assay buffer to the blank wells and 100 ng of histone extract with assay buffer to the sample wells. Plate also has wells for positive control wherein added 1 µL of purified histone provided in the kit along with the assay buffer. Incubated the plate at 37 °C for 120 min. Removed the reaction solution and washed three times with the wash buffer provided in the kit. Added detection antibody provided in the kit to each well and incubated the plate at room temperature for 60 min. Removed the detection antibody solution and washed 5 times with the wash buffer. Developer solution was added to each well and incubated in dark at room temperature for 10 min. Reaction was stopped by adding stop solution to each well and the absorbance was recorded using a spectrometer at 450 nm with an optional reference wavelength of 655 nm.

- Histone Modification Calculation:
  a. Calculated the average duplicate readings for the sample wells, assay control wells and blank wells.
  Formula 1:
  S is the amount of input sample protein in ng (used 100 ng).

$$\text{H3 Modification or total H3 (ng/µg protein)} = \frac{(\text{Sample OD} - \text{Blank OD}) \div S \times 100}{(\text{Assay Control OD} - \text{Blank OD}) \div P}$$

  P is the amount of input assay control in ng (used 25 ng).
- To calculate the percentage of histone H3 modification in total H3 using Formula 2 shown below:

$$\text{H3 Modification \%} = \frac{\text{Amount of H3 modification (ng/µg protein)} \times 100}{\text{Amount of total H3 (ng/µg protein)}}$$

## Statistical analysis

All experiments were performed at least five times independently. Data are presented as mean ± standard deviation (SD). Statistical analyses were performed using GraphPad Prism version 9. Comparisons between two groups were evaluated using either two-tailed unpaired $t$-tests or paired t-test (wherever applicable), while comparisons among more than two groups were performed using one-way ANOVA followed by appropriate post hoc tests. A $p < 0.05$ was considered statistically significant.

## Data availability

All data generated or analyzed during this study are included in this published article and its Appendix file. The original datasets have been deposited in the NCBI Gene Expression Omnibus (GEO) repository under the following accession numbers: Human transcriptomic analysis (Fig. 2)—PRJNA1305185, Rat transcriptomic analysis (Fig. 5B)—PRJNA1305751, and Rat 16S metagenomic analysis (Fig. 6E)—PRJNA1305217. Any additional information required to reanalyze the data presented in this study is available from the lead contact upon reasonable request.

The source data of this paper are collected in the following database record: biostudies:S-SCDT-10_1038-S44321-026-00403-x.

---

**The paper explained**

**Problem**

Urban particulate matter, especially fine and ultrafine particles, constitutes a major environmental threat linked to compromised pregnancy outcomes. Despite recognizing inflammation as a key biological response to UPM exposure, the mechanisms by which particle-triggered inflammatory cascades disrupt normal placental physiology and alter developmental trajectories during gestation and beyond, require comprehensive investigation. Understanding these biological pathways is fundamental to demonstrating how environmental toxin exposure translates into persistent developmental health effects.

**Results**

Our research reveals that gestational exposure to UPM activates inflammatory pathways that inhibit IGFBP3 expression, a key protein governing placental equilibrium and embryonic growth regulation. This reduction in IGFBP3 impairs critical placental processes, including trophoblast migration, metabolic exchange across the maternal-fetal interface, and vascular adaptation in uterine arteries, resulting in restricted fetal growth and altered developmental trajectories. Crucially, these molecular disruptions have lasting consequences, as prenatally exposed rodents show persistent behavioral deficits after birth, demonstrating how maternal pollution exposure mechanistically influences offspring health across the lifespan.

**Impact**

These discoveries reveal how environmental pollutants encountered early in development can change crucial biological processes through inflammation, creating long-lasting health risks. Our identification of IGFBP3 as a critical factor mediating pollution's effects on placental function creates a biological framework explaining how maternal exposures affect both fetal and child health. This mechanistic understanding opens opportunities for targeted therapeutic approaches while highlighting the vital importance of environmental protection during pregnancy to improve birth outcomes and reduce pollution-related health burdens across populations.

# Peer review information

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

## Acknowledgements

We gratefully acknowledge the Department of Biochemistry and the Animal House at the All India Institute of Medical Sciences for providing the facilities and logistical support essential for this work. We also thank the SAIF facility at AIIMS for assistance with electron microscopy. Our sincere thanks go to the Dabur Research Foundation for their invaluable support in conducting animal experiments. We are also thankful to ImageJ software as we have used it for analyzing the band intensities (Schneider et al, 2012). Graphical abstract was created using BioRender.com. SK want to thank the Indian Council of Medical Research (ICMR) and Department of Health Research (DHR), India for funding this study by grant 68.10.2023(SK).NCD-II and R.11012/03/2020-HR. SS express his gratitude to Council for Scientific and Industrial Research (CSIR), Indian Council of Medical Research (ICMR) and Department of Biotechnology (DBT) for providing the fellowship support. RD expresses her gratitude to Department of Health Research (DHR) and Indian Council of Medical Research (ICMR) for the fellowship support by grant 68.10.2023(SK).NCD-II. The funders had no role in study design, data collection and interpretation, or the decision to submit the work for publication. Institutional ethical clearance was obtained vide letter # IECPG-499/30.06.2022, dated: July 01, 2022 and F.02/IAEC/CAF/2021-2022, dated: Oct. 10, 2022.

## Author contributions

**Sunil Singh**: Data curation; Formal analysis; Validation; Investigation; Visualization; Methodology; Writing—original draft; Project administration. **Isha Goel**: Software; Formal analysis; Visualization; Methodology; Writing—original draft. **Anubhuti Rana**: Resources; Data curation; AR provided the placental samples and the pregnancy cohort database. **Anamta Gul**: Data curation; Validation. **Javed A Quadri**: Resources; Formal analysis; Investigation. **Asit Ranjan Mridha**: Resources; Formal analysis; Supervision. **Lakshay Malhotra**: Software; Formal analysis. **Neha Kashyap**: Resources; Project administration. **Baburajan Radha**: Resources; Validation. **Arnab Nayek**: Resources; Software. **Swati Ajmeriya**: Resources; Software. **Jitender Prasad**: Resources. **Ruby Dhar**: Resources; Data curation; Writing—original draft; Project administration; Writing—review and editing. **Subhradip Karmakar**: Conceptualization; Resources; Software; Formal analysis; Supervision; Funding acquisition; Visualization; Writing—original draft; Project administration; Writing—review and editing.

Source data underlying figure panels in this paper may have individual authorship assigned. Where available, figure panel/source data authorship is listed in the following database record: biostudies:S-SCDT-10_1038-S44321-026-00403-x.

## Disclosure and competing interests statement

The authors declare no competing interests.

# Expanded View Figures

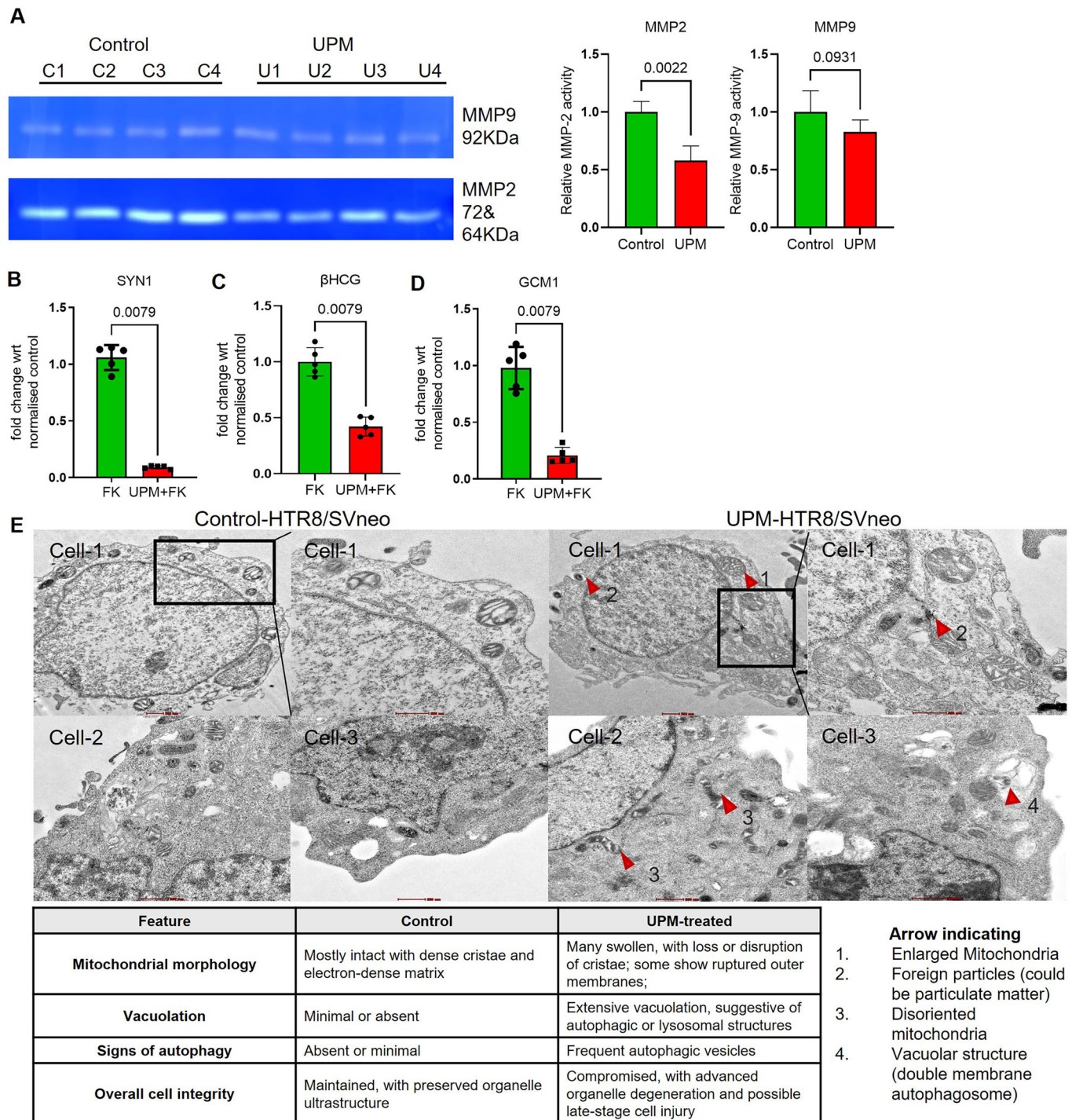

**Figure EV1. UPM exposure disrupts trophoblast invasive enzymes, syncytialization markers, and cellular ultrastructure.**

(A) Gelatin zymography analysis showing a marked reduction in MMP2 and MMP9 gelatinolytic activity in UPM-treated HTR8/SVneo cells compared with untreated controls ($n = 4$). L: ladder; C1–C4: Control replicates; U1–U4: UPM-exposed replicates. Band intensities were quantified by densitometry and normalized to total protein load. Data are presented as mean ± SD and analyzed using two-tailed Mann-Whitney test. (B–D) RT–qPCR analysis demonstrating significant downregulation of *Syncytin-1*, *β-hCG*, and *GCM1* mRNA levels in BeWo cells following UPM stimulation in the presence of forskolin ($n = 5$). Fold change values ($2^{-\Delta\Delta Ct}$) were normalized to GAPDH expression. Data are presented as mean ± SD and analyzed using two-tailed Mann-Whitney test. (E) Transmission electron microscopy (TEM) of HTR8/SVneo cells ($n = 3$) showing pronounced ultrastructural alterations following UPM exposure. Control cells displayed intact mitochondrial cristae, minimal vacuolation, and preserved organelle integrity. In contrast, UPM-treated cells exhibited swollen mitochondria, disrupted cristae, extensive cytoplasmic vacuolation, and frequent autophagic vesicles (red arrowheads). A comparative summary of morphological features between control and UPM-treated cells is provided. Scale bar equal 500 nm.

A

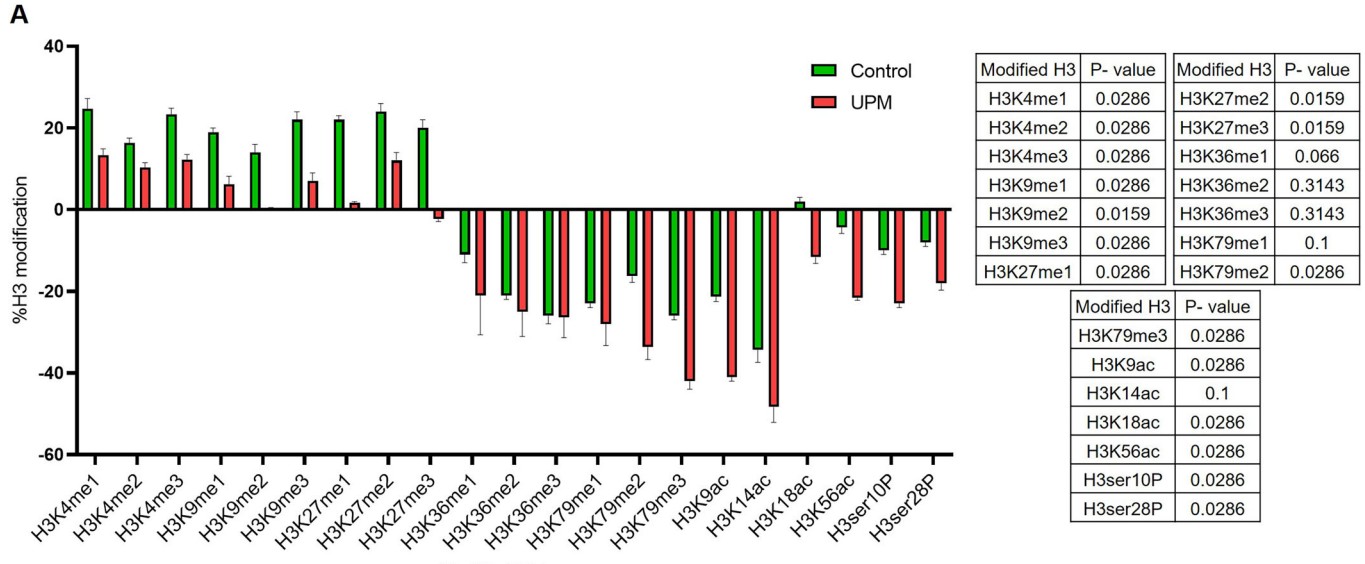

| Modified H3 | P- value | Modified H3 | P- value |
|---|---|---|---|
| H3K4me1 | 0.0286 | H3K27me2 | 0.0159 |
| H3K4me2 | 0.0286 | H3K27me3 | 0.0159 |
| H3K4me3 | 0.0286 | H3K36me1 | 0.066 |
| H3K9me1 | 0.0286 | H3K36me2 | 0.3143 |
| H3K9me2 | 0.0159 | H3K36me3 | 0.3143 |
| H3K9me3 | 0.0286 | H3K79me1 | 0.1 |
| H3K27me1 | 0.0286 | H3K79me2 | 0.0286 |

| Modified H3 | P- value |
|---|---|
| H3K79me3 | 0.0286 |
| H3K9ac | 0.0286 |
| H3K14ac | 0.1 |
| H3K18ac | 0.0286 |
| H3K56ac | 0.0286 |
| H3ser10P | 0.0286 |
| H3ser28P | 0.0286 |

Figure EV2. Global Histone H3 modification profiling.

(A) UPM exposure induced substantial alterations in the global histone H3 modification landscape within HTR8/SVneo cells. Comprehensive analysis utilizing a Histone H3 Epigenetic Modification ELISA Panel quantified 21 distinct H3 modifications in nuclear extracts from control and UPM-treated trophoblast cells ($n = 4$). Comparative bar graph visualization displayed relative changes, with green bars representing control conditions and red bars indicating UPM treatment effects. Multiple histone modifications, encompassing both methylation and acetylation marks at critical lysine residues, exhibited significant upregulation or downregulation following UPM exposure. These findings demonstrate that urban particulate matter fundamentally disrupts chromatin modification patterns and potentially mediates epigenetic reprogramming of trophoblast cellular function. Data are presented as mean ± SD. Statistical significance was determined using an unpaired two-tailed Student's t-test.

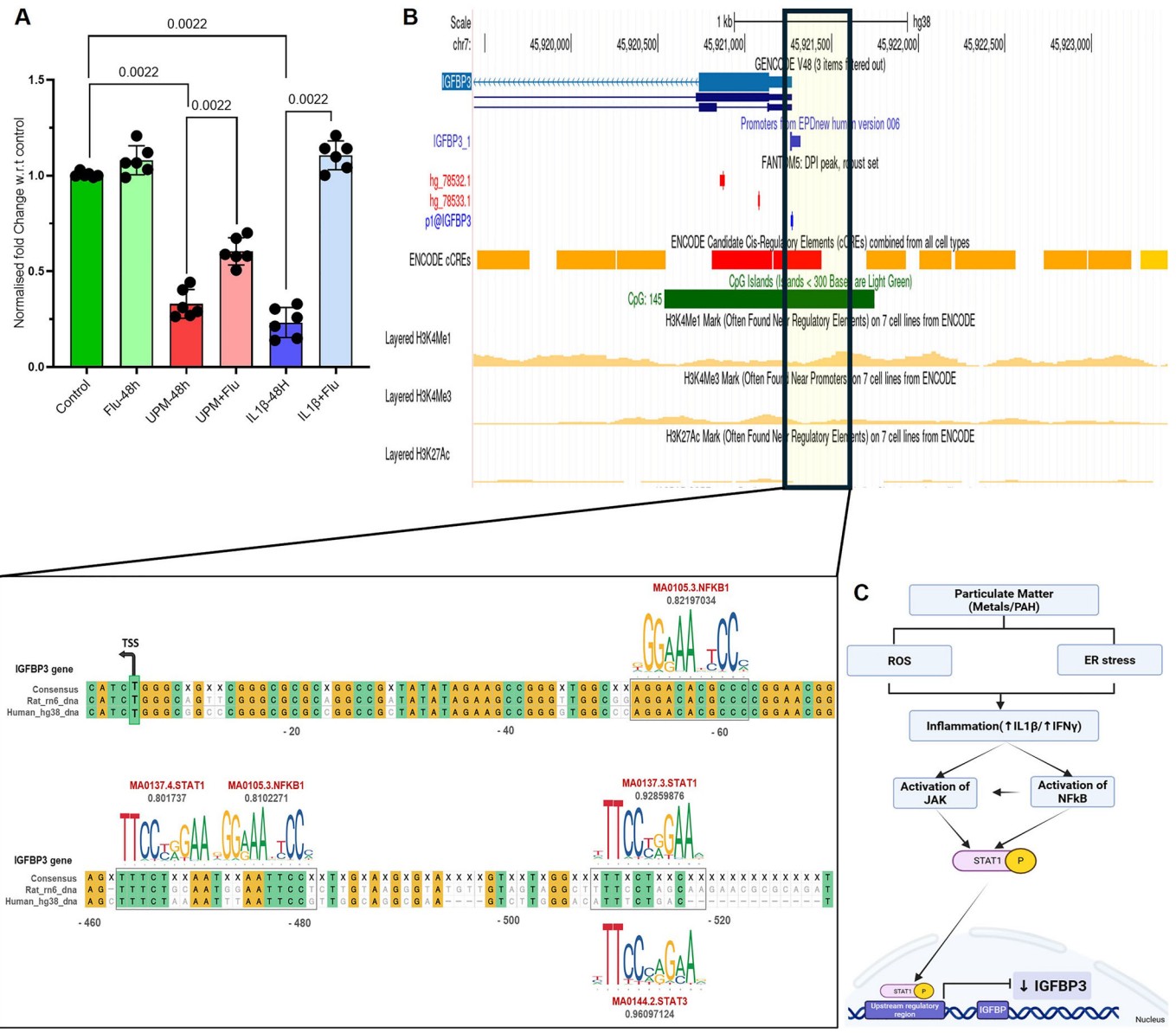

**Figure EV3. Fludarabine prevents IGFBP3 repression and an in silico promoter analysis reveals conserved STAT1/NF-κB regulatory motifs.**

(**A**) Effect of 48 h Fludarabine (Flu) (STAT1 inhibitor) on IGFBP3 expression under UPM and IL1β exposure: HTR8/SVneo cells were treated with Fludarabine (10 μM, 48 h) in the presence or absence of urban particulate matter (UPM; 50 μg/mL) or recombinant IL1β (10 ng/mL). QPCR analysis demonstrated that UPM and IL1β exposure significantly downregulated IGFBP3 mRNA levels, whereas Fludarabine treatment prevented this repression, restoring IGFBP3 expression close to basal levels. Normalized fold change in IGFBP3 expression was calculated using the $2^{-\Delta\Delta Ct}$ method, with GAPDH serving as the internal housekeeping control. Data represents mean ± SD from six independent experiments ($n = 6$). Statistical analysis was performed using one-way ANOVA. (**B**) Predicted transcription factor binding motifs in IGFBP3 promoter architecture: Inflammatory-responsive IGFBP3 transcriptional control appears to be orchestrated via STAT1 and NF-κB regulatory pathways, given the presence of computationally identified cognate binding sequences within the proximal 1 kb promoter region. This upstream regulatory domain encompasses CpG methylation sites and additional cis-acting elements. Cross-species sequence alignment reveals high conservation of STAT1 and NF-κB recognition motifs between human and rodent genomes, with corresponding binding affinity scores provided for each predicted regulatory site. (**C**) Schematic illustrating how particulate matter (containing metals and PAHs) induces ROS generation, membrane disruption, and ER stress, leading to heightened inflammatory signaling (↑ IL-1β, ↑IFN-γ). This inflammatory milieu activates JAK/STAT and NF-κB pathways, resulting in STAT1 phosphorylation and transcriptional repression of IGFBP3. The integrated pathway highlights the mechanistic link between PM exposure and impaired IGF signaling.

